# Large-scale assessment of rainfall-induced landslide hazard based on hydrometeorological information: application to Partenio Massif (Italy)

Daniel Camilo Roman Quintero[1], Pasquale Marino[1], Abdullah Abdullah[1], Giovanni Francesco Santonastaso[1], Roberto Greco[1]

[1]Dipartimento di ingegneria, Università degli Studi della Campania "Luigi Vanvitelli", via Roma 9, Aversa (CE), 81038, Italy;

*Correspondence to*: Pasquale Marino (pasquale.marino1@unicampania.it)

**Abstract**. The definition of reliable tools for rainfall-induced landslide hazard assessment is often limited by the lack of long records of landslides and relevant hydrometeorological variables. This is the case of the mountainous areas of Southern
Apennines of Campania (Italy), diffusely covered by loose pyroclastic deposits laying upon limestone bedrock, and frequently subjected to rainfall-triggered shallow landslides. To get around this issue, a 500-year long synthetic dataset of the response to precipitation of a typical slope of the area has been generated, by means of a physically based model previously validated through experimental data. The obtained dataset, containing hourly values of soil moisture and suction, and of water level in an ephemeral aquifer developing in the uppermost fractured bedrock, has been used to assess slope stability through the
calculation of the factor of safety. Based on the synthetic data, empirical thresholds for the prediction of landslide occurrence have been defined, either meteorological (i.e., based on rainfall intensity and duration) or hydrometeorological (i.e., coupling rainfall depth with antecedent root-zone soil moisture or aquifer water level). The results show that, where meteorological forcing and slope characteristics are known without uncertainty, hydrometeorological thresholds outperform the meteorological ones, and that a 3D threshold based on root-zone soil moisture, aquifer level, and rainfall depth, provides nearly
unerring landslide predictions. The use of two antecedent hydrologic variables also allows identifying two different landslide triggering mechanisms, respectively typical of the beginning and of the end of the rainy season.

To extend the application to large areas, the uncertainties linked to the spatial variability of slope geomorphologic characteristics and hydrometeorological variables were considered as random errors. Hence, foreseeing the application to the north-facing side of Partenio Massif (about 80 km$^2$), the synthetic dataset has been perturbed, superimposing Normal-
distributed random fluctuations to the calculated values of the factor of safety, and to the hydrometeorological variables used as landslide predictors. Although the uncertainty reduces the predictive skill of all the thresholds, the hydrometeorological ones show more robustness, with small numbers of both missed and false alarms. This result is confirmed by the application of the obtained thresholds to available data of landslides, rainfall and root-zone soil moisture for the period 1999-2020 in the area. The proposed methodology is an example of how to deal with uncertainty in hydrometeorological information in landslide
hazard assessment. Provided that the major hydrological processes and variables are identified, it is suitable to application also in other contexts.

**Keywords:** Hydrometeorological thresholds; Early warning; Uncertainty analysis; Conditional landslide probability.

## 1 Introduction

Rainfall-induced landslides are a common natural hazard, involving the displacement of land portions in sloping areas following heavy rain periods. The effects of landslides are critical due to human casualties, significant damages to man-made structures (e.g., roads and buildings), and substantial economic losses (Froude and Petley, 2018), and they are exacerbated by climate change and unplanned urban development (e.g., Ozturk et al., 2022). Hence, predicting the occurrence of such kind of phenomena is highly relevant and needs to be deeply investigated.

Predicting rainfall-induced landslides often relies on defining empirical thresholds separating likely non-triggering from likely triggering conditions. Empirical thresholds are indeed key components of landslide early warning systems (LEWSs). The most common threshold is the purely meteorological relationship between rainfall intensity and duration (Guzzetti et al., 2007, 2008; Segoni et al., 2018). However, it lacks physical basis and overrates data correlations, giving rise to important uncertainties, limiting the effectiveness of predictions.

It is well recognized that many rainfall-induced landslides are triggered on steep slopes covered with shallow granular deposits, usually in unsaturated conditions, following an increase in pore water pressure or a decrease of suction (Terzaghi, 1943), in turn linked to an increase in water stored in the soil (Bogaard and Greco, 2016). In the case of shallow landslides, the attainment of instability depends not only on the triggering rainfall event characteristics, but it is also favoured by antecedent wet soil conditions (Mirus et al., 2018a; Wicki et al., 2020). More generally, the accumulation of water in a slope, up to the eventual triggering of a landslide, requires that slope drainage mechanisms are not capable of effectively draining out much of the infiltrating rainwater (Bogaard and Greco, 2016; Greco et al., 2021; Marino et al., 2021). Drainage processes are controlled by the hydraulic behaviour of the boundaries of the slope, often not static, as it may change in response to various large-scale processes (in time and space) affecting the slopes as parts of larger hydrological systems. The whole processes controlling slope infiltration and drainage represent the hydrologic predisposing factors (causes), which should be considered, together with rain event characteristics (trigger), in landslide prediction (Bogaard and Greco, 2018).

In the last decade, new advancements have been made in landslide hazard research. Novel hydrometeorological thresholds that combine hydrologic predisposing factors with rainfall events leading to slope failure have been developed for landslide forecasting. Specifically, adding hydrologic information physically linked to the predisposing processes occurring in the slope (e.g., soil moisture content) has been shown to improve landslide hazard assessment from local/basin to regional scales (Abraham et al., 2021; Baum and Godt, 2010; Marino et al., 2020b; Mirus et al., 2018b; Palazzolo et al., 2023; Thomas et al., 2019). The acquisition of information about soil moisture is nowadays feasible, thanks to remote sensing (Beck et al., 2021), meteorological reanalysis products (Muñoz-Sabater et al., 2021), and through conventional field monitoring stations, at relatively low cost and in an up-to-date manner (Marino et al., 2020a, 2023). However, depending on site-specific characteristics, other hydrological variables as well as soil moisture may add valuable information to understand the seasonally changing slope response to precipitation (Illien et al., 2021; Roman Quintero et al., 2023).

Whatever is the chosen approach (i.e., purely meteorological or hydrometeorological), a common issue in the definition of statistically significant empirical thresholds is the small number of unambiguously identified (in time and space) landslide events in the historical records (Peres and Cancelliere, 2021). This is especially true when the analysis is restricted to small areas (e.g., Gonzalez et al., 2023), such as single slopes or small catchments. Expanding the studied area (i.e., moving from local to regional landslide hazard assessment) allows increasing the number of valid landslide data, but it often implies

encompassing quite different geomorphological and meteorological contexts in the same dataset, thus hampering the physical significance and reliability of the defined thresholds (Gonzalez et al., 2023; Segoni et al., 2018). One potential approach to address this issue is defining landslide thresholds based only on non-triggering conditions, much more numerous in any given dataset (Peres and Cancelliere, 2021). The dataset available for the definition of the threshold can also be enriched through the stochastic generation of synthetic data, an established technique in hydrology (e.g., Hanson, 1982; De Michele et al., 2005;

Salas et al., 2006; Keskin et al., 2006), and recently applied to landslide studies by coupling the stochastic generation of rainfall series with physically based infiltration and slope stability models (e.g., Peres and Cancelliere, 2014; Peres et al., 2018; Marino et al., 2021).

As an example of how a synthetic dataset can be exploited for landslide hazard assessment, this study refers to the case of a relatively large landslide-prone area on the north-facing slopes of Partenio Massif (Campania, Southern Italy). The study area

shares many major characteristics of landslide-affected areas in Campania: from a humid Mediterranean climate with densely vegetated slopes to soil deposits consisting of granular materials originating from the explosive eruptions of the Phlegrean Fields and the Somma-Vesuvius volcanoes (Rolandi et al., 1998). The shallow rainfall-induced landslides in the area always involve a mantle of a few meters of shallow air-fall pyroclastic deposits, mainly ashes with layers of pumices, overlaying a fractured karstic limestone bedrock, which during the rainy season hosts perched aquifers in its uppermost weathered part

(epikarst) (Greco et al., 2018).

It is well-known that the triggering mechanism leading to rainfall-induced landslides on these slopes is the reduction of soil suction in the initially unsaturated deposits, and consequently of their shear strength, caused by soil wetting during rainwater infiltration (Damiano and Olivares, 2010; Olivares and Picarelli, 2003; Pagano et al., 2010; Pirone et al., 2015). Nevertheless, many features remain unknown in the application of this knowledge to large areas especially for early warning purposes. As a

matter of fact, heavy and persistent rainfall events are sometimes followed by the triggering of landslides, but not all slopes (apparently similar) in relatively homogeneous geomorphological settings fail during the same event (Greco et al., 2021). In fact, the actual triggering of landslides depends on local geomorphologic slope features (Di Crescenzo and Santo, 2005; Crosta and Dal Negro, 2003), in some cases even affecting the mechanisms leading to slope failure. In some cases, capillary barriers develop at the ash-pumice interface and cause flow diversion and local moisture accumulation (Mancarella et al., 2012). In

others, the role of the soil-bedrock interface has been highlighted (Reder et al., 2017), which may locally impede soil drainage, favouring pore pressure build-up (Damiano et al., 2012; Damiano and Olivares, 2010), or induce soil saturation by local exfiltration from the bedrock (Cascini et al., 2008).

This paper aims at defining empirical thresholds accounting for the effects of hydrometeorological and geomechanical spatial variability for operational use in a warning system for landslide forecasting in a large area involving similar geomorphological settings. Two different approaches are compared for threshold definitions: a purely meteorological approach directly relating rainfall characteristics to landslide triggering, and a hydrometeorological approach based on a cause-trigger relationship. Both antecedent root-zone soil moisture and perched aquifer water level have been tested as possible proxies of hydrologic predisposing conditions. To obtain a dataset rich enough for statistical analyses, synthetic data are generated coupling a rainfall stochastic model with a physically based model of unsaturated flow and slope stability. Initially, the analysis is conducted for a simplified reference slope with constant inclination, covered by a homogeneous soil layer with constant thickness.

The results show how, for the reference slope, perfectly described by the physically based model used for the generation of the synthetic dataset, a 2D hydrometeorological threshold, based on soil moisture and rainfall depth, ensures very high predictive performance (i.e., no missed alarms and about one false alarm every five years). The purely meteorological I-D threshold, instead, would lead to the unacceptable rate of one false alarm per year. Furthermore, a nearly unerring 3D hydrometeorological threshold is defined, coupling perched aquifer level and root-zone soil moisture, one hour prior to the onset of rainfall, with total event rainfall depth. The three hydrometeorological variables also allow identifying the antecedent conditions leading to two different landslide triggering mechanisms.

The scale of application of the thresholds is then enlarged to the entire north-facing side of the Partenio Massif, introducing the effects of the uncertainty related to spatial variability of hydrometeorological and geomorphological variables as random fluctuations around the synthetic data. The obtained results show the robustness of the hydrometeorological thresholds, compared to the purely meteorological one. In fact, the reduction of predictive performance due to uncertainty at large scale results small, making the hydrometeorological thresholds still a reliable tool for landslide hazard assessment in the area, as confirmed by the application to available landslide data of the period 1999-2020.

## 2 Materials and methods

In this study, an area of about 80 km$^2$, the Partenio Massif, will be referred to as "large", although this term usually refers to much wider zones of regional landslide warning tools. In fact, an area with this extension includes hundreds of "small" slopes, and operational landslide forecasting relies on hydrometeorological information, typically available in few and sometimes sparse positions (i.e., the nodes of the grid of modelling or remote sensing products, or the locations of measurement instruments in the field). However, landslide triggering on the slopes of the area is influenced by local factors, (e.g., related to spatial variability of soil properties, slope and bedrock morphology, soil cover thickness and layering, vegetation, and precipitation). Additionally, information coming from monitoring or modelling nodes is affected by the spatial variability of hydrometeorological variables (e.g., rainfall, soil moisture, and aquifer level). Hence, the resulting predictive uncertainty should be considered.

In this respect, the methodology described hereinafter first analyses a single reference slope, with simplified geometry and properties known with little uncertainty, defining empirical landslide thresholds linking hydrometeorological predictors to the factor of safety of the slope. Then, the results are extended to the slopes of the large area, introducing the effects of uncertainty as random fluctuations of both the predictors and the factor of safety.

## 2.1 Case study

The study deals with the case of an area of the Southern Apennines in Campania, Italy. Specifically, the studied slopes belong to the north-facing part of the Partenio Massif, about 40 km northeast of the city of Naples, with a total extension of about 80 km$^2$ (Fig. 1). The area is the northern part of the Camp-3 warning zone defined by the Civil Protection for the management of hydro-meteorological risk in Campania (Zhang et al., 2025). They are covered by thin deposits of loose coarse-grained pyroclastic soil, consisting of alternating layers of ashes and pumices, lying upon a densely fractured limestone bedrock. The thickness of the pyroclastic deposit, typically related to slope steepness, ranges between a few meters for gentle slopes to a couple of meters around 40° inclination angle, and it tends to disappear for slopes steeper than 50°, where the underlying bedrock emerges (De Vita et al., 2006).

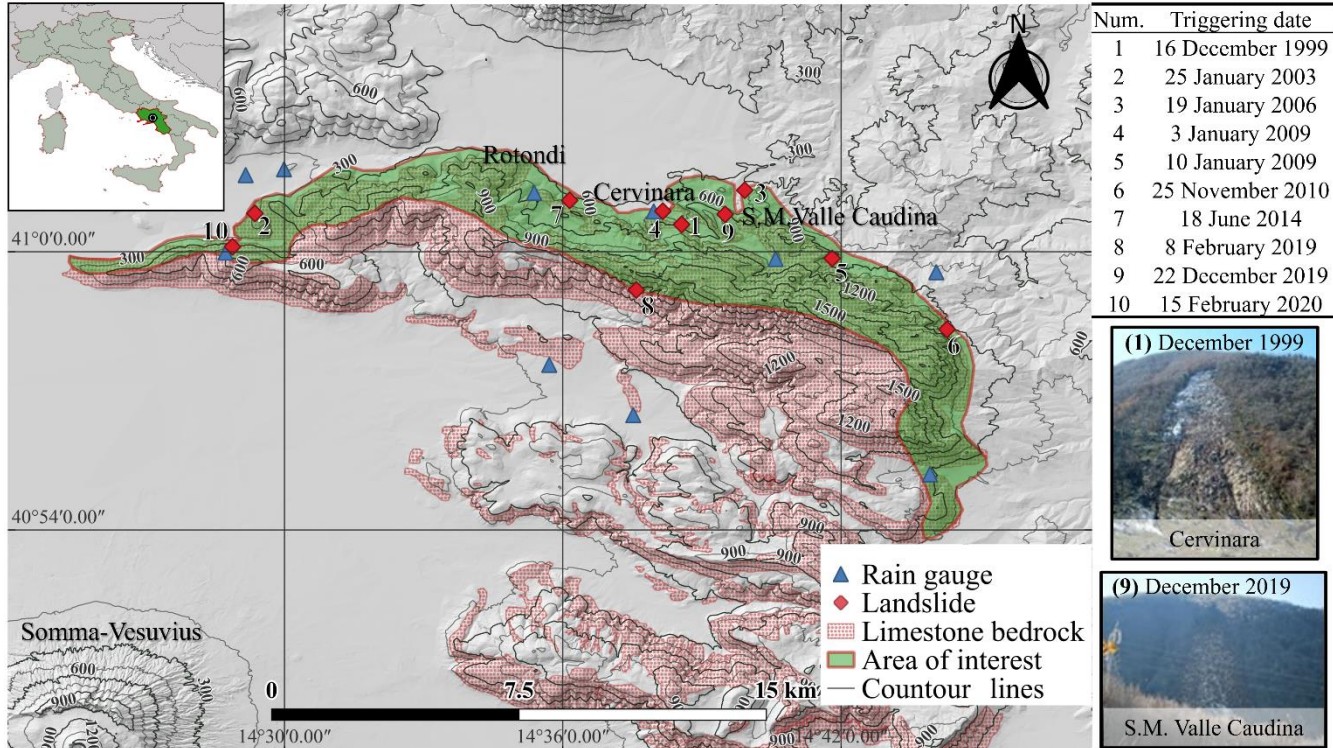

**Figure 1. Study area of north-facing part of the Partenio Massif (green-filled area) with indication of the rain gauges and the major landslides reported in different landslide catalogues from 1999 to 2020 (sourced from:** Peruccacci et al., 2023; Calvello and Pecoraro, 2018**).**

During the night between 15 and 16 December 1999, the slopes of Mt. Cornito (nearby the city of Cervinara) were affected by a series of shallow landslides after a rainfall event of about 320 mm in 48 hours. One of the major landslides moved approximately 2 km from the source area and reached the town of Cervinara (Fiorillo et al., 2001), with about 30000 m$^3$ of mobilized material, causing casualties and destroying buildings. Lately, in 2019, a debris avalanche with about 15000 m$^3$

volume affected the community of San Martino Valle Caudina (very close to Cervinara, Fig. 1), showing that this local area of a few km$^2$ belonging to northeast-facing slopes of Partenio Mountains is recurrently subjected to such kind of phenomena (Greco et al., 2021). The sliding surface of the previously described landslides, as well as in other similar geomorphological contexts of the region, usually occurs within the layers of ashes, a cohesionless soil characterized by friction angle values ranging between 37° and 39° (Olivares and Picarelli, 2003; Roman Quintero et al., 2024).

The fractured limestone formations are often the host rock for karstic aquifers. A distinction can be done between the weathered uppermost part of the bedrock, known as epikarst (Hartmann et al., 2014; Williams, 2008), more porous and pervious, and the deepest part, characterized by wider fracture systems, where the deep groundwater circulation is hosted. The interaction between the epikarst and the surface system was shown to be important for the hydrological behaviour of the unsaturated soil cover (Roman Quintero et al., 2023). The water that leaks from the soil to the epikarst forms ephemeral perched shallow

aquifers, that favour the recharge of the deep aquifer and supply surface water circulation through ephemeral springs.

## 2.2 Modelling the reference slope response to precipitation

A 500-year synthetic dataset was produced to mimic the major hydrological processes in the studied area. As is often the case with landslide studies, statistical analysis relying solely on historical observations would be limited by data scarcity. Hence, synthetic data generation is a suitable method to study slope hydrologic processes over a timescale long enough to allow for

the occurrence of slope instability multiple times (e.g., Peres and Cancelliere, 2014; Peres et al., 2018; Marino et al., 2020b). The first step in generating a synthetic dataset is defining a reliable model of the major hydrological processes occurring within a slope, which will be used to assess slope stability. In this regard, a previously developed physically based model of unsaturated flow in the soil covering a simplified slope with known geometric, hydraulic, and geotechnical characteristics (referred to hereinafter as the reference slope model) was adopted. This model should be regarded as representative of the

features of the typical slopes of the area. Although the soil cover is layered, effective parameters of a homogeneous soil layer resembling the mean hydraulic behaviour of the deposit have been assumed (Greco et al., 2013, 2018; Marino et al., 2020b). The model had been calibrated and validated according to field monitoring data and laboratory analyses, proving to be able to reliably reproduce field measurements (Comegna et al., 2016; Greco et al., 2013, 2014, 2018). The thin soil deposit in comparison to the length of the slope allows the assumption of 1D flow. The interaction with the atmosphere considers rainfall

infiltration and evapotranspiration, and the underlying perched aquifer, modelled as a linear reservoir, is connected to the overlying unsaturated soil through a coupling hydraulic condition assumed at the soil-bedrock interface (Fig. 2).

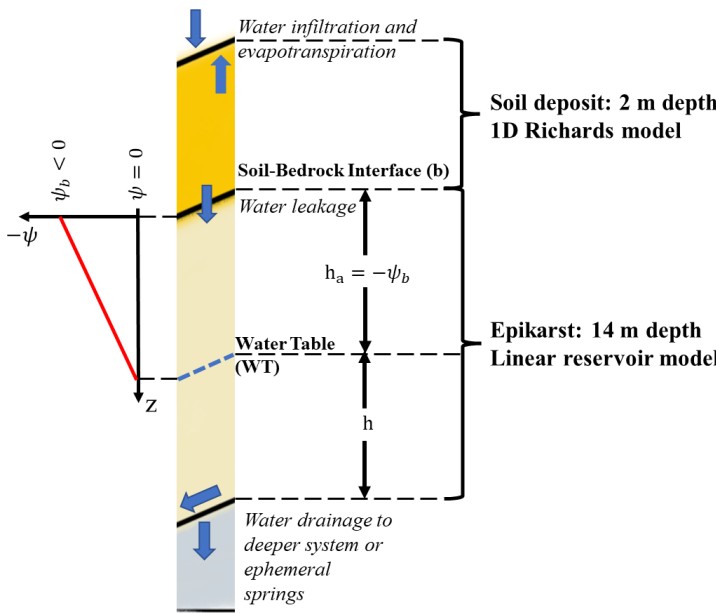

**Figure 2. Water flow scheme for the 1D coupled flow model considering (from top to bottom): rainwater infiltration from the soil surface, unsaturated flow through the soil deposit accounting the effect of evapotranspiration, water leakage through the soil-bedrock interface, and saturated water flow in the epikarst joined to the soil-bedrock interface affecting the water potential.**

The water flow in the unsaturated soil deposit is modelled assuming a homogeneous soil layer, using the 1D Richards' equation (1), where $\psi$ is the soil matric potential, $\theta$ the volumetric water content, $z$ is the vertical direction (see Fig. 2), and $k$ is the unsaturated hydraulic conductivity. The actual evapotranspiration is considered in the source term $q_r$ in Eq. (1), representing root water uptake, estimated from the potential evapotranspiration with the model of (Feddes et al., 1976), assuming a triangular root distribution penetrating the entire cover thickness.

$$\frac{\partial \theta}{\partial t} = \frac{dk}{d\theta}\frac{\partial \theta}{\partial z} + \frac{\partial}{\partial z}\left(k\frac{d\psi}{d\theta}\frac{\partial \theta}{\partial z}\right) - q_r \qquad (1)$$

The hydraulic characteristic curves, $\theta(\psi)$ and $k(\psi)$, are modelled with the van Genuchten-Mualem model shown in Eqs. (2) and (3), where $\theta_r$ and $\theta_s$ are the residual and saturated volumetric water content, respectively; $\alpha$, $n$, and $m = 1 - 1/n$ are shape parameters; $S_e = (\theta - \theta_r)/(\theta_s - \theta_r)$ is the effective degree of saturation; $K_s$ is the saturated hydraulic conductivity (Mualem, 1976).

$$\theta(\psi) = \begin{cases} \theta_r + \dfrac{\theta_s - \theta_r}{(1 + \alpha|\psi|^n)^m} & \text{if } \psi < 0 \\ \theta_s & \text{if } \psi \geq 0 \end{cases} \qquad (2)$$

$$k(\psi) = \begin{cases} K_s S_e^{0.5}\left[1 - \left(1 - S_e^{1/m}\right)^m\right]^2 & \text{if } \psi < 0 \\ K_s & \text{if } \psi \geq 0 \end{cases} \qquad (3)$$

The unsaturated flow model in the soil cover is coupled with a linear reservoir model simulating the water accumulation in the epikarst aquifer. The water balance equation of the perched epikarst aquifer reads:

$$n_a \frac{dh}{dt} = i_b - \frac{h}{K_a} \qquad (4)$$

In Equation (4), $n_a$ is epikarst effective porosity, h is the water table depth, $i_b$ is the water leakage from the soil cover to the bedrock, and $K_a$ is the time constant of the linear reservoir. The aquifer water level is assumed to linearly affect the suction ($\psi_b$) at the base of the soil deposit, $\psi_b = H_e - h$, where $H_e$ is the epikarst thickness. All model parameters are summarised in Table 1.

**Table 1. Hydraulic parameters applied in the coupled model of 1D water flow for the unsaturated soil cover and for the perched aquifer hosted in the Epikarst.**

|  |  |  |
|---|---|---|
| **Soil cover** | $\boldsymbol{\theta_s}$ (-) | 0.70 |
|  | $\theta_r$ (-) | 0.01 |
|  | $\alpha$ (m$^{-1}$) | 6 |
|  | $n$ (-) | 1.3 |
|  | Soil cover thickness (m) | 2 |
|  | Saturated hydraulic conductivity, $K_s$ (m·s$^{-1}$) | 3x10$^{-5}$ |
|  | Slope inclination angle, $\beta$ (°) | 40 |
| **Epikarst** | Epikarst thickness, $H_e$ (m) | 14 |
|  | Effective porosity (-) | 0.005 |
|  | Time constant of linear reservoir (days) | 871 days |

The above-described model was run to simulate the water flow with a 500-year synthetic hourly rainfall series, generated with the Neyman-Scott Rectangular Pulse model (NSRP) (Rodriguez-Iturbe et al., 1987). The NSRP is a stochastic model that reproduces the rainfall process based on a random selection of the beginning, rainfall amount and duration of single rainfall cells (rectangular pulses), that can overlap with each other. Specifically, every cell width and height represent the duration and the intensity of each rainfall cell, respectively. Hence, when many cells overlap, the total intensity at any time is the result of the direct summation of the intensities of the overlapping cells. The model has been calibrated with the method of moments (Cowpertwait et al., 1996; Peres and Cancelliere, 2014), based on a 17-year long rainfall dataset with a time resolution of 10 minutes, from the rain gauge of Cervinara, managed by the Civil Protection Agency of Campania (Marino et al., 2020b; Roman Quintero et al., 2023).

Equations (1) and (4) have been solved with the finite differences technique in a 2 cm spacing grid with hourly timestep, allowing to obtain the 500-year synthetic series of volumetric water content $\theta$ and water potential in the soil profile, and of perched aquifer water level in the epikarst (hereinafter it will be indicated as the depth of the water table below the soil-bedrock interface, $h_a = -\psi_b$, as sketched in Fig. 2).

Based on the results of the model simulations, slope stability is assessed by evaluating the factor of safety (FS) at every simulated time. The assumed 1D geometry allows carrying out the slope stability analysis under the infinite slope hypothesis, i.e., FS being the ratio between the resistive shear strength and the effective shear stress, derived from the equilibrium analysis of a soil column element of height $d$ resting on a slope with inclination angle $\beta$:

$$FS = \frac{c' + \gamma d \cos^2 \beta \tan \phi' - \gamma_w S_e \psi \tan \phi'}{\gamma d \sin \beta \cos \beta} \qquad (5)$$

In Equation (5), $c'$ and $\phi'$ are effective soil cohesion and friction angle, respectively; $\gamma_w$ is the unit weight of water; $\gamma$ is the mean unit weight of the wet soil column; $\chi$ is the Bishop coefficient, function of $\psi$ and here assumed equal to the effective degree of saturation ($\chi = S_e$) (Lu and Likos, 2006). As a layer of cohesive altered ashes is present near the soil-bedrock interface, with a typical thickness of about 50 cm, the failure depth is assumed at the base of the cohesionless soil profile, i.e., 1.5 m below the ground surface (Marino et al., 2021). Landslide triggering is assumed to occur whenever FS < 1.0.

Once the landslide triggering has been identified, an association between the rain events and the occurrence or non-occurrence of landslides is needed. To this aim, rainfall events have been separated within the 500-year hourly rainfall series, assuming that rainfall can be regarded as a new event only when the effects of the previous one have disappeared from the topsoil, the moisture of which controls rainwater infiltration during the new event. To this aim, a separation interval of 24 hours with rain depth smaller than the mean daily evapotranspiration, estimated as 2 mm/day, has been assumed, as after this time the topsoil moisture returned to the field capacity for all the simulated rainfall events (Roman Quintero et al., 2023). Under this assumption, 26363 rain events were identified, with durations between 1 and 427 h and total rainfall depth between 2 mm and 645 mm.

The way the rain events have been associated with the occurrence or non-occurrence of landslides is shown in Fig. 3.

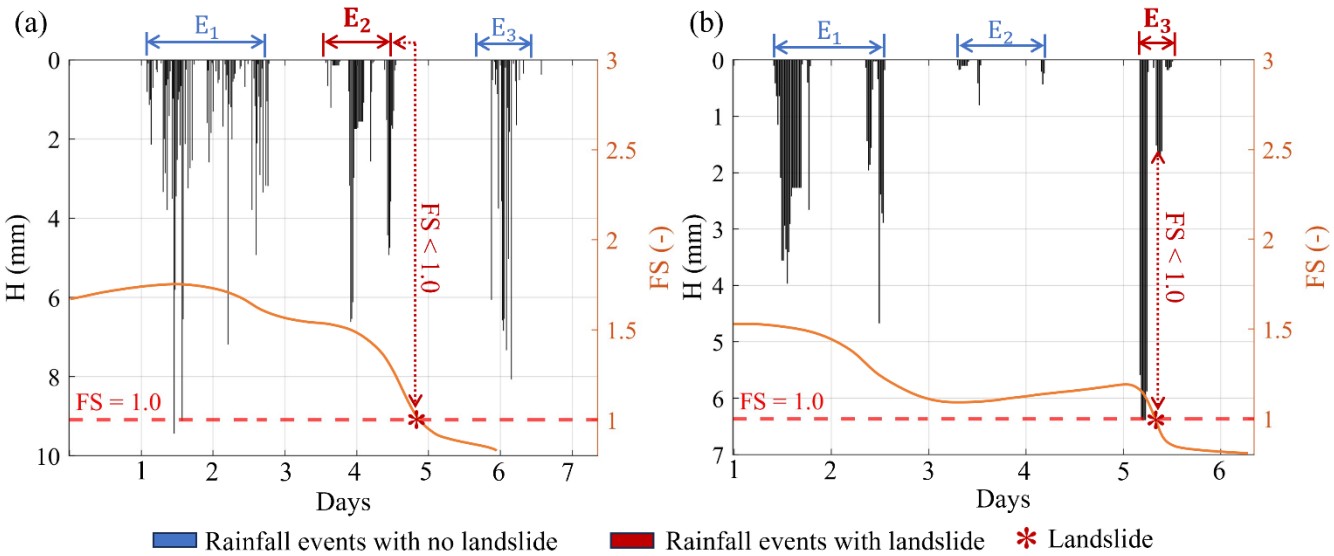

**Figure 3. Schematic representation of the occurrence of landslides associated with rainfall events in two different cases: (a) associated with the previous event if FS < 1.0 after the end of the rainfall; (b) associated with the actual ongoing event if FS < 1.0 within the rainfall event.**

Within the time series, the landslide occurrence condition (i.e., FS < 1.0) might be related to the triggering rain event in two different ways, as depicted in Figs. 3a and 3b, which show three rainfall events, $E_1$, $E_2$ and $E_3$. In the first case (Fig. 3a), FS < 1.0 is attained during a dry interval, owing to the delay in rainwater infiltration, thus the triggering of the landslide is associated with the previous rainfall event ($E_2$). In the second case (Fig. 3b), the critical condition is attained during the rainfall event $E_3$, and so the landslide is associated with the ongoing event. To avoid misinterpretation of the actual triggering event, whenever the rainfall event associated with a landslide resulted smaller than 20 mm, it was merged with the immediately previous one.

## 2.3 Empirical landslide threshold definition

Empirical thresholds are a useful tool for separating rainfall events with landslides from those without. They consist in a line, or a surface, plotted in a suitable 2D or 3D space, often adopted in landslide early warning systems. The definition of empirical thresholds can be made with different functional formats, according to the shape of the data cloud in the chosen space (Mirus et al., 2024).

The hydrologic analysis of the synthetic dataset carried out by (Roman Quintero et al., 2023) has showed that, in the studied geomorphological context, the fraction of water remaining stored in the soil cover at the end of a rainfall event ($\Delta S/H$) is strongly related to the mean volumetric water content of the uppermost 100 cm of the soil profile ($\theta$) and the antecedent perched aquifer water level ($h_a$), both evaluated one hour before the initiation of the event. This suggests that these two variables may be used for the definition of hydrometeorological landslide thresholds. Hence, different empirical thresholds have been here tested, aiming at comparing their landslide forecasting performances, ranging from the traditional meteorological rainfall intensity-duration threshold ($D, I$) to the hydrometeorological one ($\theta, H$), in which the hydrological information θ has been considered as representative of conditions that predispose the slopes to failure, to novel functional format for hydrometeorological threshold defined in the 3D space ($\theta, h_a, H$).

The parameters of all the tested functional formats for the empirical thresholds have been identified by maximizing the True Skill Statistic TSS (Peirce, 1884), which gives a measure of the predictive performance of the threshold:

$$\text{TSS} = 1 - \frac{M}{P} - \frac{F}{N} \tag{6}$$

In equation (6), $M$ is the number of missed alarms (i.e., events not exceeding the threshold, but followed by a landslide), $P$ is the total number of landslides (true positives), $F$ is the number of false positives (i.e., rainfall events exceeding the threshold, without any landslide occurrence) and $N$ is the total number of rainfall events not followed by any landslide (true negatives). A perfectly working threshold curve gives TSS = 1, while TSS = −1 indicates an always failing threshold. The objective function (6) is optimized by using a Genetic Algorithm (GA) (Goldberg and Holland, 1988).

Basically, it is worth to note that whenever $N \gg P$ (i.e., landslide triggering is a rare phenomenon), thus TSS is more sensitive to the missed alarms $M$, rather than to the false alarms $F$. Nevertheless, this bias can be useful for landslide prediction purposes, because a missed alarm error could have catastrophic consequences, while false alarms may only cause inconvenience to the

involved inhabitants and may affect their responsiveness and trust in an early warning system in the long run, owing to the well-known cry-wolf effect (Breznitz, 1984).

## 2.4 Real slope response to precipitation

Compared to the reference slope, the real slopes of the study area present variable characteristics, in terms of geomorphology as well as hydraulic and mechanical properties of the soil. Spatial variability of slope characteristics, as well as of rainfall input, affects both the assessment of slope stability and the representativeness of the values of the variables used for the definition of the empirical threshold.

In this respect, to mimic the operational use of the empirical thresholds at large scale (i.e., referred to north-facing part of the Partenio Massif, about 80 km$^2$), the effects of the uncertainty affecting all the variables should be considered. Specifically, Normal-distributed fluctuations are superimposed on the synthetic meteorological and hydrological variables used for the definition of the local thresholds.

### 2.4.1 Uncertainty in the hydrometeorological variables for threshold definition for a large area

In the reference slope, both the meteorological (i.e., rain event duration, depth, and mean intensity) and hydrological variables (i.e., root zone soil moisture and perched aquifer water level), that can be used for empirical thresholds definition, are perfectly representative of the actual meteorological forcing and hydrologic antecedent conditions of the slope. Instead, in real operational applications to a wide area, the same variables would be the result of sparse measurements or of model simulations with limited spatial resolution. Hence, the characteristics of the considered rainfall events are not those affecting all the slopes of the area. Similarly, the values of antecedent hydrological variables would not remain unchanged if measured (or simulated) at different points of the study area. Therefore, uncertainty affects the hydrometeorological variables to be used for threshold application to a large area. The greatness of uncertainty is related to the extension of the area to which the threshold is applied, compared to the spatial density of the input data.

To gain insight into rainfall uncertainty in the study area, the spatial variability of rainfall amount of events, separated as described in section 2.2, has been analyzed. Rainfall recorded in the period 2001-2021 by three rain gauges managed by the Civil Protection in Cervinara, Rotondi, and San Martino Valle Caudina (S.M.V.C.) has been used (Fig. 1). The distance between the rain gauges ranges from 4.8 km (between Rotondi and Cervinara) to 10.0 km (between Rotondi and S.M.V.C.), with the rain gauge of S.M.V.C. installed at a significantly higher altitude compared to the other two (Table 2). The rain events observed at different rain gauges, that overlapped for at least one hour in the time series, were considered contemporary. When the different intermittency of the rain observed at two gauges gave rise to the correspondence of a single event at the first gauge with multiple shorter events at the second, the short events were merged in a single event, so to obtain comparable total rainfall depths (i.e., in these cases, the separation criterion of 24 hours with less than 2 mm was disapplied). Table 2 reports some statistical indices of the difference of the rainfall observed at the three gauges, including the total number of events at each gauge, and the number of events overlapping with any of the others.

**Table 2. Characteristics and degree of overlapping of rainfall events recorded at the three rain gauges of the study area.**

| Rain gauge | Elevation (m a.s.l.) | Mean yearly rainfall (mm) | Minimum and maximum event depth (mm) | Minimum and maximum event duration (h) | Total number of events | Number of events overlapping with Cervinara | Number of events overlapping with Rotondi | Number of events overlapping with S.M.V.C. |
|---|---|---|---|---|---|---|---|---|
| Cervinara | 349 | 1600 | [2, 266.6] | [1, 219] | 1010 | 1010 | 845 | 831 |
| Rotondi | 483 | 1500 | [2, 260.6] | [1, 218] | 1010 | 845 | 1010 | 840 |
| S.M.V.C. | 850 | 2000 | [2, 403] | [1, 290] | 1052 | 831 | 840 | 1052 |

As an example, for two couplings of rain gauges, i.e. Cervinara with Rotondi and Cervinara with San Martino Valle Caudina, the scatterplots of the differences, $\Delta H = H_1 - H_2$, vs. the mean values, $H_{med} = (H_1 + H_2)/2$, of event depths recorded at the two gauges are reported in Fig. 4. The coupling between Rotondi and San Martino Valle Caudina has not been represented, as it resulted very similar to that of Cervinara with San Martino Valle Caudina.

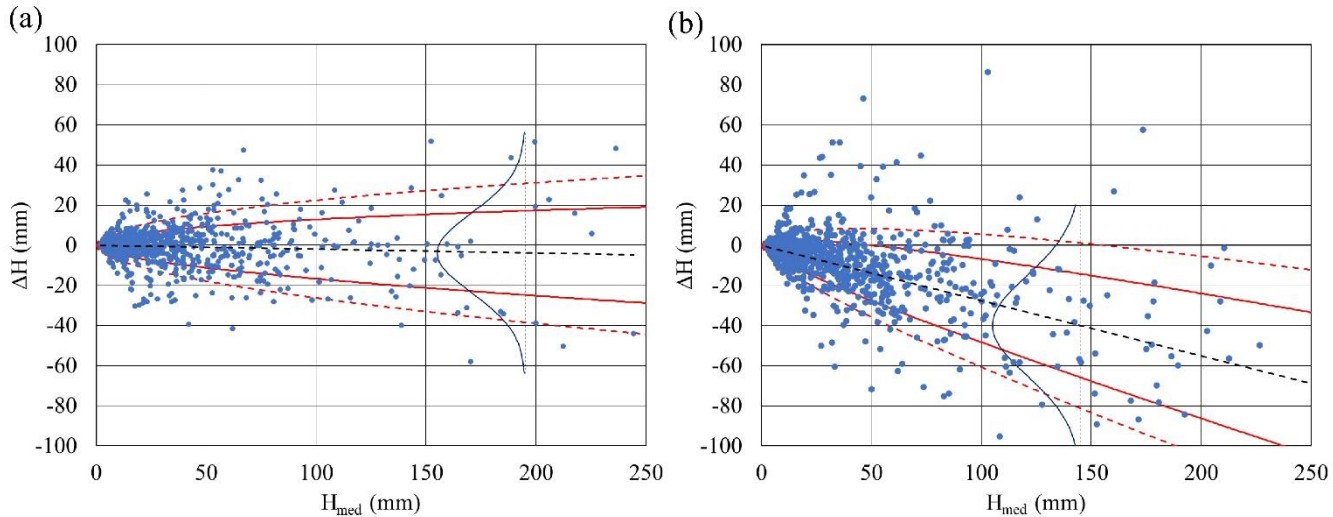

**Figure 4. Scatterplots of the differences $\Delta H$ vs. the mean values $H_{med}$ of rainfall depths of events recorded at the rain gauges of (a) Cervinara and Rotondi, and (b) Cervinara and San Martino Valle Caudina. The black dashed line represents the linear fitting of the $\Delta H(H_{med})$ relationship. The solid and dashed red lines include 68.3% and 86.6% of the dots, respectively. The blue bell-shaped curves represent the identified Normal distributions.**

The black dashed line in each graph represents the linear fitting of the dependence of the fluctuation $\Delta H$ on $H_{med}$, i.e., $\Delta H = $
$\alpha H_{med}$. The solid and dashed red lines, symmetric around the black dotted line, represent equations of the format $\Delta H = \alpha H_{med} \pm A \times H_{med}^B$, with the power-law terms characterized by the same exponent $B$ and different values of the coefficient $A$. The exponent $B$ and the coefficients $A$ of the solid and dashed curves, reported in table 3, have been obtained by searching the curves delimiting the zones containing 68.3% and 86.6% of the dots, respectively, and leaving outside the same number of

dots above and below them. If the ratio of the coefficients *A* obtained for the dashed and solid lines is close to 1.5, this indicates

that the dot clouds are arranged in such a way to allow assuming the fluctuations of the rain depth difference to be Normal-distributed around their mean, with mean and standard deviations depending on the mean total event depth according to the above mentioned linear and power-law relationships, respectively.

**Table 3. Dependence of mean and standard deviation of rain event total depth fluctuations on mean total depth, for coupled rain gauge stations of the north side of Partenio mountains.**

| First rain gauge ($H_1$) | Second rain gauge ($H_2$) | $\alpha$ | A (68.3%) | A (86.6%) | B |
|---|---|---|---|---|---|
| Cervinara | Rotondi | -0.020 | 1.30 | 2.15 | 0.53 |
| Cervinara | San Martino Valle Caudina | -0.276 | 1.38 | 2.21 | 0.59 |


The analysis of rain events for the three considered stations shows that both the distance and the difference in altitude affect the spatial variability of rainfall depth. Specifically, the rain event depths recorded at the stations of Rotondi and Cervinara, located at the foot of the north-facing slopes at close altitude, share the same mean value, and the spreading of the depth difference around the mean looks similar. Whereas the rain gauge of San Martino Valle Caudina, at a significantly higher

altitude, is considered, the dependence of mean event rain depth on the altitude clearly arises, and the spreading of rain depth difference looks slightly larger, as if the difference of elevation also affects event depth fluctuations and not only their mean value. However, the analysis of how topographic factors affect the spatial variability of rainfall depth in the Partenio Massif cannot be based on the analysis of only three gauges, but it should be more deeply investigated based on rain data from all the rain gauges operating in the study area.

Regarding the variability of rain event depth moving away from the gauge position, the analyses for the three considered rain gauges indicate that the standard deviation of rain event depth at about 5 km from the measurement point can be approximated as $\sigma(H) \cong 1.5 \times H^{0.5}$. However, the density of the rain gauges managed by the Civil Protection agency in the study area (Fig. 1) is such that each rain gauge covers an area of about 10 km$^2$, so that the maximum distance of a slope from the closest rain gauge can be considered smaller than 2 km. Hence, to introduce the uncertainty deriving from the spatial variability of rain,

the following relationship has been assumed for the standard deviation of rain event depth:

$$\sigma(H) = 0.75 \times H^{0.5} \tag{7}$$

Moreover, the uncertainty on the hydrological variables $\theta$ and $h_a$ was introduced by considering a Normal-distributed random Absolute Error (AE) with zero mean and known standard deviation. Firstly, in the case of $\theta$, the standard deviation of the AE distribution was assumed to be equal to 0.02. Brocca et al. (2012) showed that in areas smaller than 100 km$^2$ monitored through

conventional stations, an AE of $\pm 0.04$ on the readings of $\theta$ can be addressed with a 95% confidence with a relatively low sampling density. Figure5a shows the distribution of the AE on $\theta$, where 95% of the data fall between approximately $\pm 0.04$. Similarly, (Dari et al., 2019) estimated that an AE in an area of about 500 km$^2$ instrumented with 12 stations shall be capable of give lectures with an error around 0.03, somehow confirming this picture even for larger areas. However, measurements obtained through remote sensing techniques, such as ERA-5 land products, often display a typical error around 0.02 in the

studied area (Hersbach et al., 2020; Muñoz-Sabater et al., 2021). Thus, the selected standard deviation of 0.02 encompasses the expected variability in volumetric water content measurements across different methodologies and environmental conditions.

Likewise, the standard deviation of the distribution of AE accounting for the uncertainty in the ground water monitoring, was assumed here to be equal to 1 m. Figure5b presents the assumed distribution of AE on $h_a$, ranging around $\pm 2$ m with a 95%
confidence level. With this assumption, the perturbed water levels will very unlikely approach the ground surface, consistently with the gushing out of water in natural springs at ground, observed rarely in the area, and only downhill, near the main streams Cornito and Castello (Marino et al., 2020b; Roman Quintero et al., 2023). The aim of introducing $h_a$ is to assess the potential use of ephemeral aquifer formation in slope areas as an indicator of the conditions affecting water accumulation in the soil cover, including active and impeded drainage mechanisms (Roman Quintero et al., 2023). Indeed, some studies indicate that
ephemeral aquifer systems can emerge in highlands and their water levels, related to wet and dry seasons, are spatially stable in wide areas (e.g., Bennett et al., 2022). In this respect, some data provided by one-year observations from wells placed downhill in the study area, show that the groundwater table depth difference at the wells seems to agree with the assumed maximum AE, ranging around $\pm 2$ m (Autorità di Bacino, 2013).

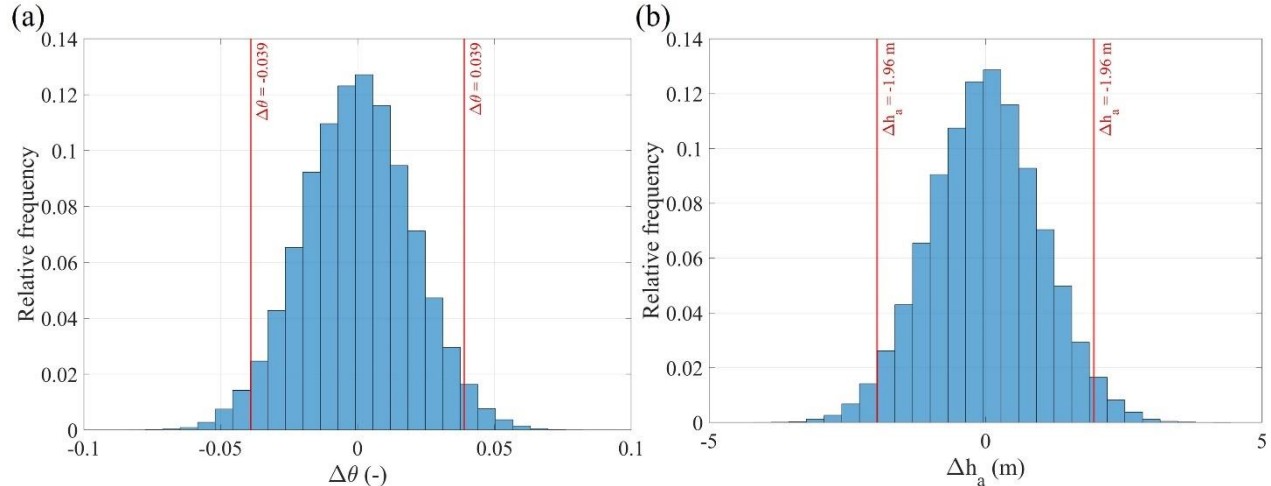

**Figure 5. Frequency distributions of the absolute error associated with (a) the volumetric water content, Δθ, and (b) the ground water level, Δh$_a$, assumed to randomly perturb the original dataset.**

### 2.4.2 Uncertainty in the assessment of slope stability

In the FS equation (5), the uncertainty of failure surface depth, $d$, and of slope inclination, $\beta$, depend on the variability of slope morphology; uncertainty of soil strength parameters, $c'$ and $\phi'$, is linked to the variability of soil properties; uncertainty of soil
column mean unit weight, $\gamma$, and of the value of $\psi$ at the considered depth, may depend on the variability of both slope morphology and soil properties, as well as on the spatial variability of the meteorological forcing (mainly rainfall), vegetation and boundary conditions. Hence, the uncertainty affecting the calculated values of FS has been assessed as the combined

effects of all these uncertain factors, characterized as Normal-distributed. The standard deviations assumed for each parameter, given in Table 4, are such to encompass the variability of the properties of the soils of the area, reported in experimental studies (e.g., Roman Quintero et al., 2024, and references therein).

**Table 4. Mean values and standard deviations of the factors affecting the uncertainty of the values of the factor of safety calculated for the simplified slope model.**

| Variable | Mean value (reference slope) | Standard deviation (large scale) |
|---|---|---|
| $d$ (m) | 1.5 | 0.1 |
| $\beta$ (°) | 40.0 | 2.0 |
| $c'$ (N/m$^2$) | 0.0 | 0.0 (always assumed cohesionless soil) |
| $\phi'$ (°) | 38.0 | 0.5 |
| $\gamma$ (N/m$^3$) | variable for every event, depending on calculated soil moisture profile | 10% of the mean |
| $\psi$ (N/m$^2$) | variable for every event, depending on calculated FS value | 1000.0 |
| $FS$ (-) | variable for every time | 0.1 |

The values of the standard deviations reported in Table 4 for the large scale can be considered representative of the spatial variability of the characteristics of the slopes affected by shallow landslides in the north-facing side of Partenio Mountains (Fig. 1). In fact, the pyroclastic soil deposits, originated by the same eruptions, share similar stratigraphy and physical properties (e.g., Roman Quintero et al., 2024), thus explaining the small uncertainty affecting $\phi'$ and $c'$ (this latter has been considered constantly equal to zero, as this is a conservative assumption for slope stability assessment). The factors related to slope morphology also present limited variability in the study area, as big landslides usually occur in a relatively narrow range of inclination angles. In fact, slopes above 45° inclination typically present a very thin soil coverage (Del Soldato et al., 2018; De Vita and Nappi, 2013), while those with inclinations smaller than 35°, due to the high effective friction angle, would fail only if pore water pressure becomes positive. However, soil saturation is very unlikely in the considered slopes, thanks to the high porosity and hydraulic conductivity of the soil, as well as to the perviousness of the underlying fractured bedrock. Thus, the standard deviation of the fluctuation of FS around the values calculated for the simplified slope model results 0.10 (Table 4).

## 3 Results

### 3.1 Reference slope

The reference slope stability analysis applied to the synthetic dataset leads to 20 triggering events in 500 years. In Fig. 6, the scatterplots of triggering (red points) and non-triggering (green points) rainfall events are shown in the plane of rainfall duration and intensity ($D$, $I$), as well as in the plane of mean volumetric water content of the uppermost 100 cm and the rainfall events height ($\theta$, $H$). In the two planes, meteorological (Fig. 6a) and hydrometeorological (Fig. 6b) thresholds have been defined with

power-law ($I = a\,D^b$) and linear ($H = a\theta + b$) functional formats, respectively, which share the same number of parameters. Table 5 summarizes the obtained parameters and the performance metrics for the tested thresholds.

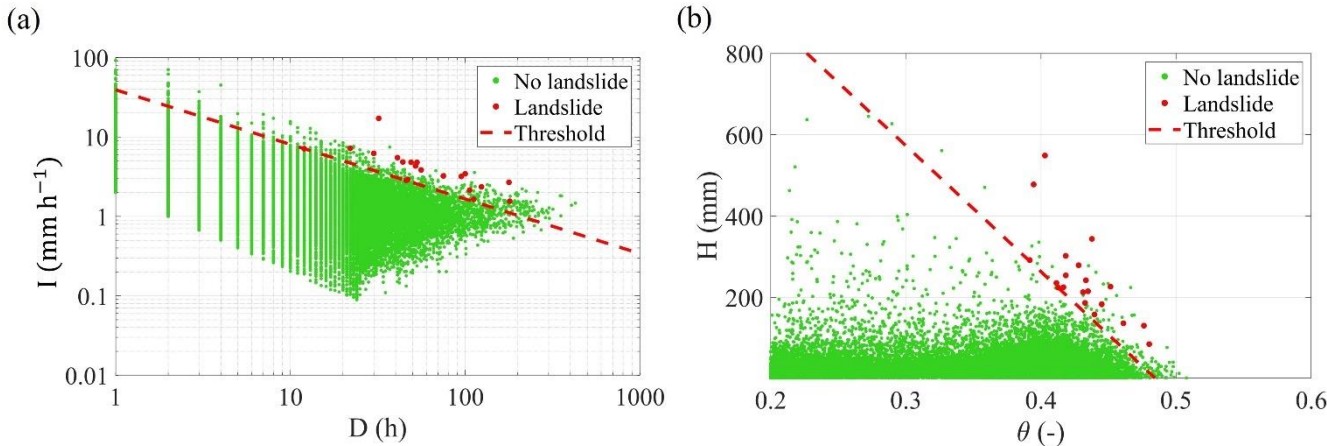

Figure 6. Power-law meteorological threshold (a); linear hydrometeorological threshold (b). In the 500 year-long synthetic dataset, the red dots represent the rainfall events followed by the triggering of a landslide; the green ones are rainfall events after which no landslide occurs.

Table 5. Parameters and metrics of the tested functional formats for the meteorological and hydrometeorological thresholds plotted in Fig. 6.

| Functional formats | A | b | TSS | Missed Alarms | False Alarms | Total events |
|---|---|---|---|---|---|---|
| Meteorological Power-law (Fig. 6a) | 39.22 | -0.69 | 0.983 | 0 | 452 | 26363 |
| Hydrometeorological Linear (Fig. 6b) | -3097.86 | 1502.76 | 0.996 | 0 | 96 | |

Both thresholds show good predictive performance, as indicated by the high values of TSS and the absence of missed alarms. However, the proposed hydrometeorological linear threshold performs better in terms of the frequency of false alarms, which decreases from about once a year to once every five years.

Moreover, the addition of a second hydrological variable ($h_a$) allows considering the influence of the conditions at the lowermost boundary of the soil deposit, as shown in Fig. 7. Specifically, a novel 3D threshold format, consisting of two planes ($\theta$, $H$) parallel to the axis $h_a$ and separated by a limit value of perched aquifer water level ($h_x$), is proposed. In particular, the bi-plane format has 5 parameters: slope and intercept of the two traces of planes in the coordinate plane ($\theta$, $H$), and the limit value of antecedent aquifer level $h_x$ (i.e., $H = a_1\theta + b_1$ with $h_a \geq h_x$ and $H = a_2\theta + b_2$ with $h_a < h_x$, respectively, Fig. 7b and Fig. 7c). The parameter values and the performance metrics obtained by maximization of TSS are reported in Table 6.

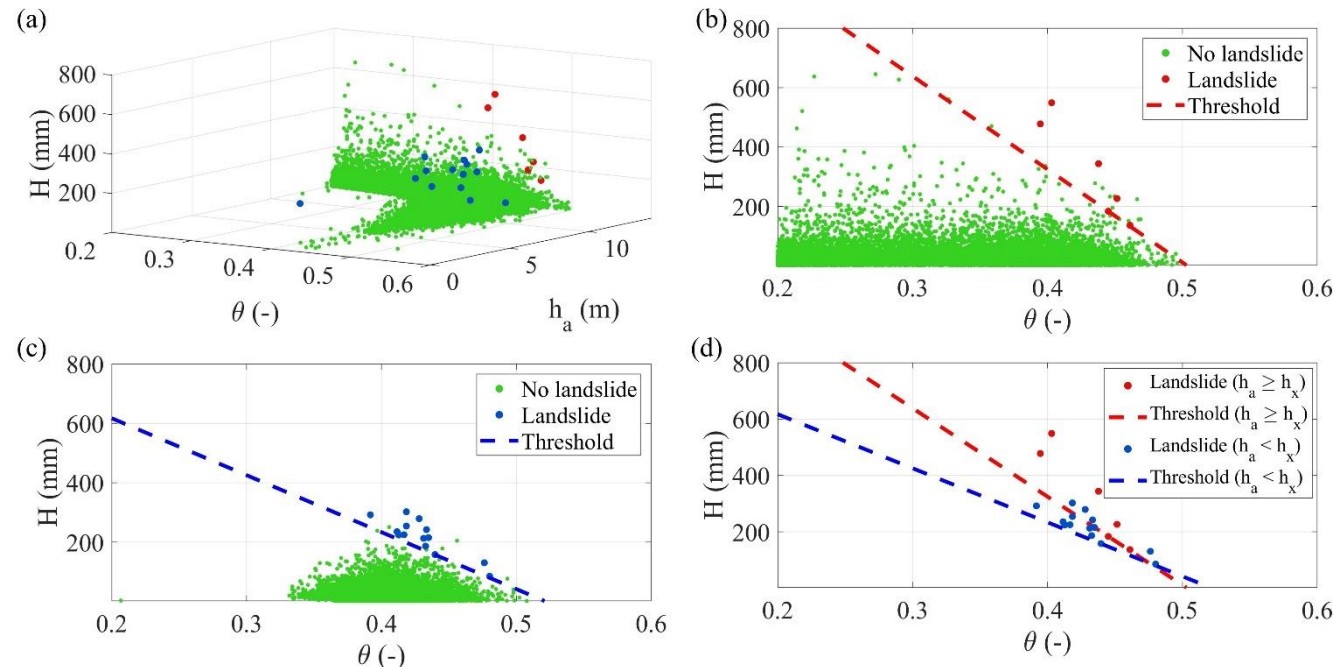


**Figure 7. Optimal hydro-meteorological threshold in the space ($\theta$, $h_a$, $H$): (a) scatter plot of data in the 3D space; (b) $h_a \geq 12.4$ m (low perched aquifer water level) and (c) $h_a < 12.4$ m (high perched aquifer water level). The green dots indicate conditions not leading to landslides. Landslide events have been split in red and blue dots, corresponding to different antecedent aquifer level; (d) comparison of the obtained thresholds for low and high aquifer level.**

**Table 6. Parameters of the tested functional format for the 3D hydrometeorological threshold defined in Fig. 7.**

| Functional format | Antecedent conditions | a | b | $h_x$ | TSS | Missed alarms | False alarms | Total events |
|---|---|---|---|---|---|---|---|---|
| Bi-plane | $h_a \geq h_x$ | -3134.03 | 1578.39 | 12.39 | 0.999 | 0 | 18 | 26363 |
| | $h_a < h_x$ | -1916.91 | 1000.42 | | | | | |

It is worth noting from Table 6 that the performance of this threshold is further improved compared to those obtained with 2D analysis (Table 5), leading to the lowest total number of false alarms, i.e., only 18 in 500 years. The obtained value $h_x$ (about 12.4 m below the ground surface) separates low perched aquifer water level (i.e., $h_a \geq h_x$), a condition typical of late autumn,

from high water level ($h_a < h_x$), observed after persistent rainy seasons.

**3.2 Large scale**

As described in section 2, aiming at introducing the effects of uncertainty on the assessment of slope stability, the series of FS has been perturbed with Normal-distributed random fluctuations, with standard deviation 0.10.

Similarly, the spatial variability of the hydrometeorological variables ($\theta$, $h_a$, $H$), used for the definition of the threshold for the

reference slope (section 3.1) and supposed to be measured at a single place for the considered area, has been introduced.

Specifically, 100 different perturbed series (hereinafter "scenarios") have been generated embedding both the hydrological ($\theta$ and $h_a$) and meteorological ($H$) information as probabilistic variables with Normal-distributed random fluctuations with zero mean and the standard deviation values reported in section 2.4.1. For the sake of simplicity, the duration and mean intensity of rainfall events have been assumed to give the same contribution to the fluctuations of $H$. Hence, the perturbations of duration and mean intensity have been obtained from the generated perturbations of $H$ assuming $1 + \Delta I/I = 1 + \Delta D/D = (1 + \Delta H/H)^{1/2}$.

For all the scenarios, Fig. 8 shows the scatter plots of the perturbed variables in the planes $(D, I)$ and $(\theta, H)$. The red dots represent the conditions followed by landslide occurrence, the green dots those without landslides. It looks clear from the dot arrangement that, even considering the randomness due to the spatial variability of all the considered variables, the hydrometeorological variables $\theta$ and $H$ separate landslides from non-landslides much more effectively than the meteorological variables $I$ and $D$.

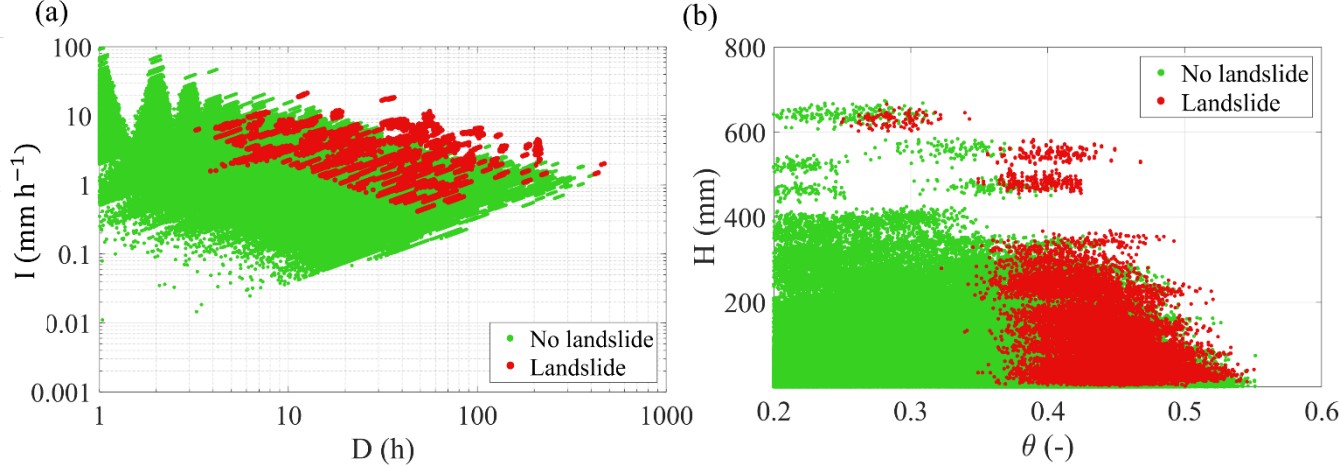

**Figure 8. Scatter plot of perturbed meteorological and hydrometeorological variables of 100 scenarios: (a) in the meteorological plane ($D$, $I$) with logarithmic axes; (b) in the hydrometeorological plane ($\theta$, $H$).**

The suitability of threshold lines has been investigated by defining an optimal threshold for each scenario (i.e., maximizing TSS), for all the functional formats tested for the reference slope. Fig. 9 shows the scatter plot of the perturbed variables with the dashed thresholds obtained by calculating the mean value of threshold line parameters obtained for all scenarios, in the $(D, I)$ plane with power-law equation (Fig. 9a), in the $(\theta, H)$ plane with linear equation (Fig. 9b), and in the 3D space $(\theta, h_a, H)$ with the bi-plane format (Figs. 9c and 9d). In the latter case, the events with landslides have been split in red and blue dots, according to the low and high-water level, respectively. Moreover, the shaded areas in the plots of Fig. 9 represent the spanning of the 100 thresholds obtained for all the scenarios.

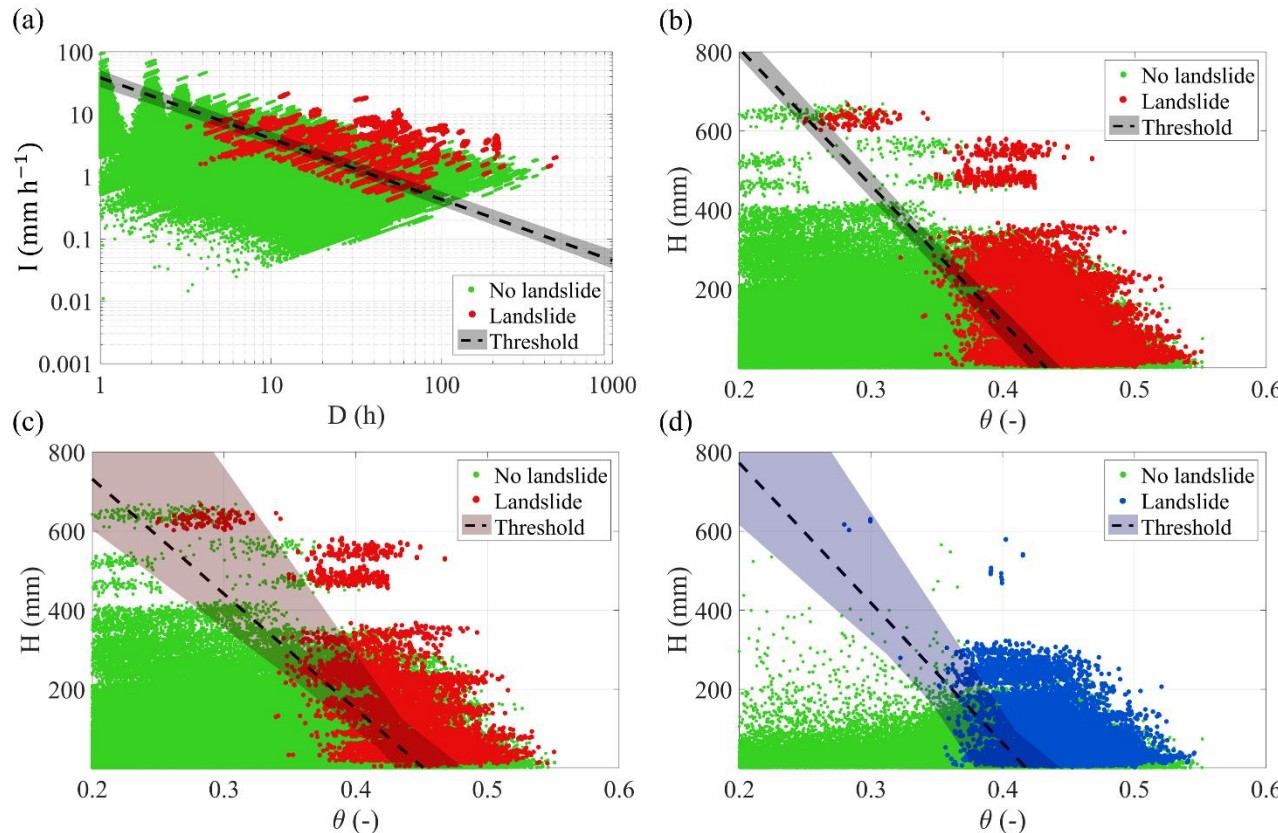

**Figure 9. Scatter plot of perturbed meteorological and hydrometeorological variables with the thresholds obtained by the optimization of 100 scenarios using different functional formats: (a) Power-law meteorological thresholds in the meteorological plane ($I, D$) with logarithmic axes; (b) linear hydrometeorological thresholds in the plane ($\theta, H$); Bi-plane hydrometeorological thresholds in the plane ($\theta, H$) according to different antecedent aquifer level $h_a$: (c) low perched aquifer water level ($h_a \geq h_x$: red dots) and (d) high perched aquifer water level ($h_a < h_x$: blue dots).**

The predictive performances displayed by the investigated thresholds for the large area (80 km²) are consistent with those obtained for the reference slope. In fact, the performance increases moving from pure meteorological, to hydrometeorological 2D, to hydrometeorological 3D, as sketched in Fig. 10, in which the distributions of the obtained TSS values for all 100 scenarios are shown through box and whiskers plots.

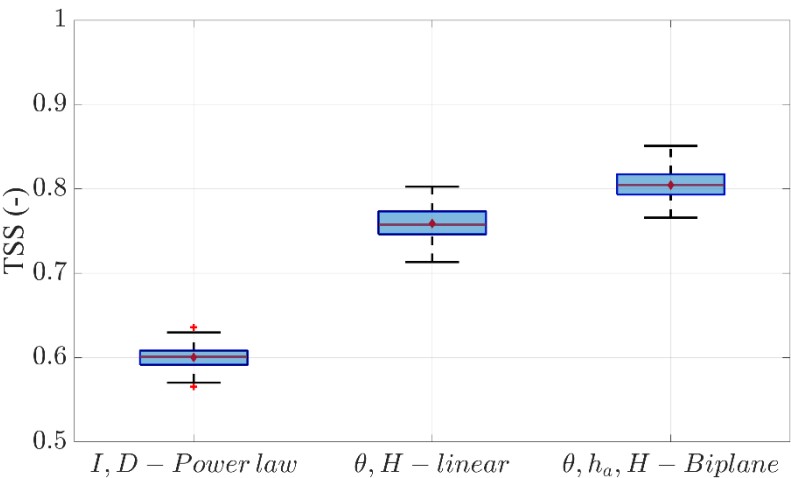

**Figure 10. Performance comparison of the tested functional formats for all scenarios referred to meteorological and hydrometeorological thresholds.**

As depicted in the box plots, the hydrometeorological approach leads to TSS values ranging between 0.72 and 0.80 (2D threshold), and between 0.77 and 0.85 (3D threshold). Instead, the meteorological approach shows TSS values between 0.57 and 0.64. For each functional format, the red lines inside the box represent the median value, overlapping with the average of TSS (red dots in the middle of the boxes). For the three tested formats, Table 7 summarizes the parameters and the performance indicators of the median TSS, as well as the overall TSS range (ΔTSS).

**Table 7. Summary of performance indicators obtained by maximizing the TSS values of the landslide thresholds applied to the different scenarios.**

| Functional formats | | a | B | $h_x$ | TSS median | Missed alarms | False alarms | Total events | ΔTSS |
|---|---|---|---|---|---|---|---|---|---|
| **Meteorological** | Power-law | 38.63 | -0.98 | - | 0.601 | 34 | 5362 | | 0.07 |
| **Hydrometeorological** | Linear | -3467.71 | 1502.54 | - | 0.757 | 16 | 3962 | 26363 | 0.08 |
| | Bi-plane | -2896.64 | 1308.85 | 11.03 | 0.804 | 11 | 3406 | | 0.08 |
| | | -3559.64 | 1478.81 | | | | | | |

It is worth noting that the median linear hydrometeorological threshold allows obtaining a total number of false alarms in the whole area equal to 3962 in 500 years (i.e., on average, 8 per year), with missed alarms once every 31 years.

Moreover, the bi-plane 3D threshold further improves the predictions, leading to the lowest total number of missed alarms, i.e., only 11 in 500 years (on average once every 45 years), as well as the lowest number of false alarms, as reported in Table 7.

The perturbed data clouds allow the estimation of the landslide probability distributions, reported in Fig. 11a and Fig. 11b in the meteorological and hydrometeorological planes, respectively. To draw the graphs, a rectangular grid with variable spacing has been defined, so to have at least 20 dots falling within each rectangle. The landslide probability has been estimated in each rectangle as the ratio between the number of landslides and the total number of dots. The shape of the isolines of probability

in the transition zone from small to high values in the two planes also confirms that the power-law equation (a straight line in a log-log plane) and the linear equation, respectively sketched in the two plots, are suitable functional formats for the threshold

lines in the $(D, I)$ and $(\theta, H)$ planes. More specifically, the transition from small to high probability of landslide looks sharper in the $(\theta, H)$ plane than in the $(D, I)$ plane (notice that this latter graph is plotted in logarithmic scales). This confirms that a threshold line separating landslides from non-landslides can ensure a higher performance in the $(\theta, H)$ plane, simultaneously limiting both missed and false alarms.

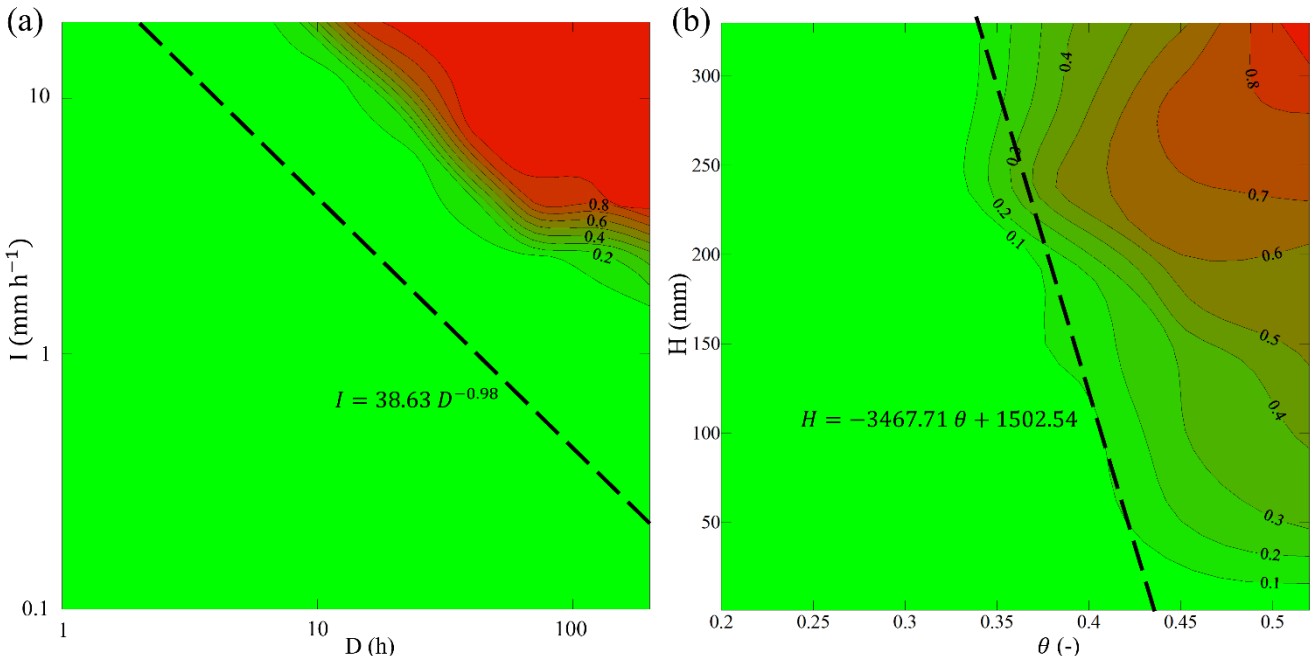

**Figure 11. Conditional landslide probability distributions: (a) in the meteorological plane (D, I) with logarithmic axes; (b) in the hydrometeorological plane $(\theta, H)$. The dashed lines represent the thresholds corresponding to the median TSS among all the perturbed scenarios.**

## 4 Discussion

The synthetic dataset reliably reproduces the response of the reference slope to precipitation. In fact, the obtained frequency of landslides (i.e., about one every 20 years) well agrees with the recurrence interval of landslides triggered on very nearby slopes (e.g., Greco et al., 2021). In this respect, the introduction of uncertainty also leads to reasonable results, with landslides occurring in the entire study area once every two to three years, in agreement with the 10 landsides reported in the entire study area in the period 1999-2020 according to existing landslide catalogs (Trigila et al., 2010; Calvello and Pecoraro, 2018;

Peruccacci et al., 2023).

The richness of the dataset allows a sound evaluation of the performance of empirical landslide thresholds, comparing different choices of predictors and functional formats. The results indicate that the predictive performance of the hydrometeorological threshold using root zone soil moisture and rainfall depth as predictors, strongly tied to the physics behind landslide triggering mechanisms, is less affected by uncertainty than the widely used meteorological threshold based on precipitation intensity and duration. Both thresholds ensure very good performance for the reference slope, with TSS close to 1. When uncertainty is introduced, the predictive performance decreases in both cases. However, the reduction of TSS is 24.1% for the hydrometeorological threshold, while it is 37.5% for the meteorological one.

Furthermore, the inclusion of the aquifer water level as a second hydrological predictor, thus proposing a 3D threshold, not only improves the predictive performance, but also sheds light on two possible landslide triggering mechanisms described in the study area. In fact, some authors suggested that slope failure can be ascribed to the wetting of the uppermost part of the soil due to intense rainwater infiltration (saturation from the top), favored by the presence of dry coarse layers within the soil profile (e.g., Mancarella et al., 2012). Other authors, instead, linked slope failure to the filling of the underlying fractured bedrock after prolonged periods of rain. This condition hampers water leakage from the soil deposit and sometimes even induces exfiltration from the bedrock into the soil deposit (saturation from the bottom) (e.g., Cascini et al., 2008). The proposed 3D threshold clearly highlights the different predisposing and triggering conditions of each of the two possible mechanisms.

Another significant result made possible by the generation of a large synthetic dataset is the probabilistic assessment of the conditions leading to landslide occurrence. Figure 11 shows that the empirical thresholds defined by maximizing the TSS are quite conservative, as they correspond to low landslide probability values. This is due to the rareness of landslides, compared to the total number of rainfall events. In the meteorological plane, where rainfall intensity and duration cannot clearly separate landslides from non-landslides, the threshold line lies in a zone where landslide probability is only around 0.03, thus implying many false alarms. The hydrometeorological predictors, instead, better describe the extreme nature of the conditions leading to landslides, with a clearer separation of the conditions that lead to landslides from those that do not. This allows drawing the optimal threshold line simultaneously limiting missed and false alarms, as confirmed by its position in the probability plane (Fig. 11b), where it lies close to the isolines of probability around 0.15.

This result highlights the importance of the process-based identification of the predictor variables, achieved thanks to the physically based modelling approach adopted. In fact, the definition of an empirical threshold for landslide initiation requires a strong simplification of the reality and may lead to good predictive performance only if the major processes controlling the response of the slope to precipitation are correctly identified (Roman Quintero et al., 2023). For the highly conductive and loose coarse-grained pyroclastic deposits of the study area, the infiltration capacity is so high that runoff rarely occurs, even during heavy rainstorms (Marino et al., 2020a). Consequently, parameters affecting runoff generation, such as rainfall intensity or topsoil moisture, result unimportant. The exponent obtained for the optimal I-D threshold, close to -1 (i.e., the threshold line corresponds to a nearly constant value of H), also confirms that the intensity of rain events plays a minor role, Rainfall depth and antecedent root zone soil moisture, instead, ensure reliable predictions, as they are directly related to the accumulation of water in the soil, required for slope failure. The inclusion of the aquifer water level as a second hydrological

predictor, accounting for the effectiveness of drainage through the soil-bedrock interface, provides a slight further improvement. It is worth noting that in a different geomorphological context, where the hydrological processes affecting slope response to precipitation might be different, the most informative hydrological predictors would be different as well, and their choice should be guided by case-specific hydrological modelling.

However, the simplicity of thresholds as tools for landslide hazard assessment inevitably leads to errors, which can be limited, but not eliminated, using suitable hydrologic predictors. The larger is the considered area, likely including slopes with variable characteristics, the larger the expected prediction errors. Therefore, the definition of the zones where a single landslide threshold is applied, as well as the design of monitoring networks supplying the required information, should rely on hydrological modelling.

## 4.1 Application to real landslide dataset

A practical application to a real dataset has been carried out for the period from 1999 to 2020, during which 10 landslide events were registered in the study area (Trigila et al., 2010; Calvello and Pecoraro, 2018; Peruccacci et al., 2023). Table 8 summarizes the major features of the reported landslides.

The rain events and the corresponding predictor variables (i.e., $H$, $D$ and $I$) have been extracted from the time series of rainfall recorded by the Cervinara rain gauge, except for the landslide that occurred in 1999, when it had not yet been installed. In total, the rainfall series contains 862 rain events.

For the root zone volumetric water content, $\theta$, the ERA5-Land reanalysis data was compared to field measurements in the pyroclastic deposit available in Cervinara (Marino et al., 2020a). The remotely sensed data reliably reproduces the trends of soil moisture measured in the field, but underestimates its range of variability, owing to the extremely high porosity of the pyroclastic soil of the area. Therefore, ERA5-Land root zone soil moisture was linearly rescaled to cover similar intervals as the field measured data. The values obtained are in good agreement with the modelled ones, and they allow mimicking the operational use of the thresholds without the support of any model.

**Table 8. Main features of landslides reported in northern Partenio Massif between 1999 and 2020.**

| Nr. | Source | Date | Spatial accuracy[4] (km²) | Temporal accuracy[5] (days) | $\beta$[6] (°) | $H$[7] (mm) | $D$[7] (hours) | $I$[7] (mm/h) | Antecedent soil moisture Modelled[8] | ERA5[9] |
|---|---|---|---|---|---|---|---|---|---|---|
| 1 | IFFI[3] | 19/12/1999 | Exact | <1 | 40 | 356.4[10] | 68[10] | 5.24[10] | 0.47 | n.a. |
| 2 | Italica[1] | 25/01/2003 | 1 | 1 | 37 | 170.4 | 106 | 1.61 | 0.42 | 0.42 |
| 3 | Italica[1] | 19/01/2006 | 100 | 1 | 48 | 28.8 | 33 | 0.87 | 0.40 | 0.37 |
| 4 | Italica[1] | 03/01/2009 | 1 | <1 | 36 | 203.6 | 65 | 3.13 | 0.41 | 0.41 |
| 5 | Italica[1] | 10/01/2009 | 1 | <1 | 41 | 45.0 | 42 | 1.07 | 0.46 | 0.46 |
| 6 | Italica[1] | 25/11/2010 | 1 | 1 | 44 | 179.2 | 111 | 1.61 | 0.44 | 0.46 |
| 7 | Italica[1] | 18/06/2014 | 1 | >1 | 34 | 146.4 | 27 | 5.42 | 0.36 | 0.30 |
| 8 | FraneItalia[2] | 08/02/2019 | n.a. | n.a. | 42 | 16.0 | 8 | 2.00 | 0.46 | 0.48 |
| 9 | IFFI[3] | 22/12/2019 | Exact | <1 | 42 | 127.2 | 54 | 2.36 | 0.45 | 0.45 |

| | | | | | | | | | | |
|---|---|---|---|---|---|---|---|---|---|---|
| 10 | FraneItalia[2] | 15/02/2020 | n.a. | n.a. | 39 | 14.8 | 7 | 2.11 | 0.36 | 0.32 |

[1]Italica landslide catalog (Perruccacci et al., 2023). [2]FraneItalia landslide catalog (Calvello and Pecoraro, 2018). [3]Italian Landslide Inventory (Trigila et al., 2010). [4]Spatial accuracy represents the area of a circle, around the reported coordinates,
where the landslide might have occurred. [5]Temporal accuracy represents the time interval, around the reported date, when the landslide might have occurred. [6]The slope inclination angle refers to the reported coordinates, regardless of the reported spatial accuracy. [7]Rainfall data from the rain gauge of Cervinara. [8]Root zone soil moisture modelled for the reference slope with the precipitation input of the Cervinara rain gauge. [9]Root zone soil moisture at the reported coordinates interpolated on the ERA5-Land grid. [10]Rainfall data from the rain gauge of S. Martino Valle Caudina of the Italian Hydrographic Service, dismissed in
560 2000.

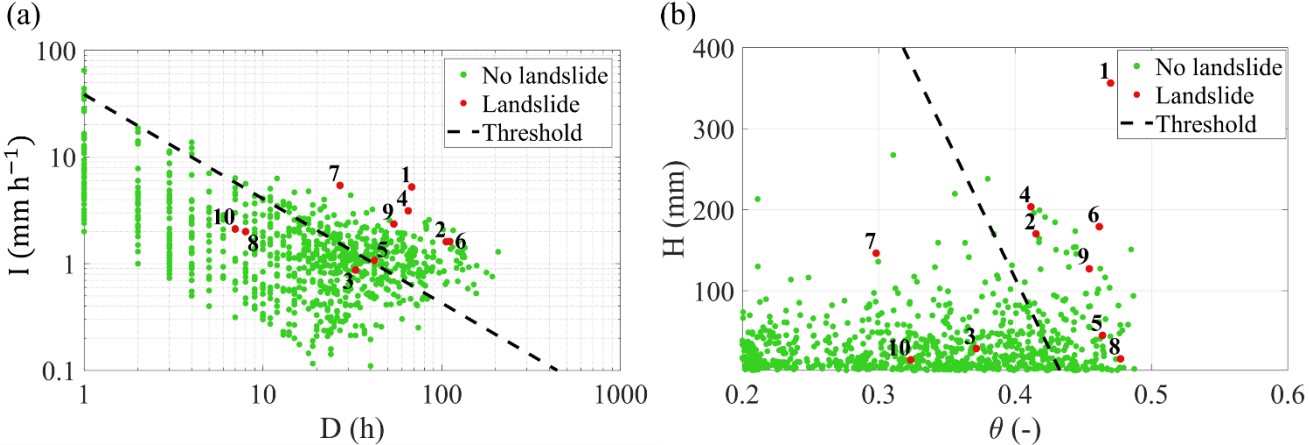

**Figure 12. Application to real landslides, from 1999 to 2020, of the power-law meteorological (a) and the linear hydrometeorological**
**thresholds (b) defined in the section 3.2. In both planes, the scattered dots representing rainfall events not followed by any landslide (green dots) and rainfall events after which landslides were registered (red dots).**

The median 2D meteorological and hydrometeorological thresholds (Table 7) have been tested for assessing their performances referred to the real landslide dataset (Fig. 12). The graphs of Figure 12a and 12b compare, respectively, the meteorological $(D, I)$ and hydrometeorological $(\theta, H)$ thresholds with the scatter plot of recorded rain events and landslides. The
hydrometeorological threshold performs better also in the prediction of real landslide occurrences. Indeed, it reaches a TSS value of 0.53, whereas for the purely meteorological it is 0.42. The number of false alarms is consistent with the results obtained with the synthetic dataset (i.e., 143 and 235 in 19 years for the hydrometeorological and meteorological thresholds, respectively), but the missed alarms, three for both the thresholds, are significantly more than expected.

However, this result can be ascribed to intrinsic uncertainties affecting the landslide inventories, where indeed some of the
reported landslides have been classified as having very limited geographic (i.e., landside nr. 3) or temporal (i.e., landside nr. 7) accuracy. Furthermore, the rain data from the gauge of Cervinara may not be suitable for some of the farthest landslides

(e.g., the reported locations of landslides nr. 2, 6 and 10, represented in Fig. 1, are more than 10 km far from the rain gauge). The correction of just some of the three missed landslides would bring the predictive performance of the thresholds back to a similar level as for the synthetic dataset.

**5 Conclusions**

The paper investigates the potential advancements that may be achieved by including one or more hydrological variables in the definition of empirical thresholds for rainfall-induced landslide forecasting. To this aim, a novel methodology based on physically based modelling is proposed. It allows identifying the major hydrological processes controlling landslide occurrence and dealing with the uncertainty introduced by the spatial variability of hydrometeorological variables. As landslide inventories
usually provide limited amounts of data suitable for statistical analyses, a 500-year long synthetic dataset is generated by means of the physically based model.

The proposed methodology is applied to the Partenio Massif, an example of the wide areas of Campania (Italy), characterised by carbonate massifs covered with a thin layer of pyroclastic deposits, frequently affected by rainfall-induced shallow landslides. The model used for the generation of the synthetic dataset had been previously calibrated and validated thanks to
detailed laboratory and field experiments. It reproduces the response to meteorological forcing of a reference slope, with known geomorphologic and physical properties.

The analysis of the synthetic dataset of the reference slope, including soil moisture and suction distributions in the soil deposit, as well as perched aquifer water level, allowed the assessment of slope stability at hourly resolution. The results show that a hydrometeorological threshold, based on root-zone soil moisture and rainfall depth, outperforms the usually adopted purely
meteorological threshold, based on rainfall intensity and duration. Furthermore, a novel hydrometeorological threshold, defined in the 3D space of root-zone soil moisture and aquifer water level (as indicators of antecedent slope conditions) and rainfall depth, provides nearly unerring predictions of landslide triggering. The proposed 3D threshold also allows identifying the antecedent conditions leading to the activation of two different landslide triggering mechanisms, related to the beginning and the end of the rainy season.

To extend the analysis to a large area, the effects of the spatial variability of slope characteristics and rainfall has been introduced as Normal-distributed random perturbations of the reference slope variables. In fact, uncertainty affects the assessment of slope stability, as well as the representativeness for a large area of hydrometeorological variables measured or modelled at few locations. The distributions of landslide probability, conditional to meteorological (i.e., rain intensity and duration) and hydrometeorological (i.e., root-zone soil moisture and rain depth) variables clearly indicate that these latter are
more robust with respect to uncertainty, as the transition from small to high landslide probability is sharper in the hydrometeorological plane. This result is confirmed by the smaller reduction of the predictive performance of the hydrometeorological threshold applied at large scale, compared to the meteorological one.

The proposed approach, although based on synthetic data, looks promising for operational early warning application, as shown by the comparison with real landslide data for the north-facing side of Partenio Massif, with an extension of about 80 km$^2$.

Specifically, rainfall data from a rain gauge of the area and root-zone soil moisture data from ERA5 meteorological reanalysis have been used to assess the performance of the obtained meteorological and hydrometeorological thresholds for the prediction of the landslides of the period 1999-2020 reported in available inventories. The small numbers of missed and false alarms indicate that the hydrometeorological threshold obtained with the synthetic data can be used as an effective tool for landslide early warning in the study area. To this aim, the possibility of getting information about perched aquifer level (e.g., linking it to measurements of water level in streams supplied by ephemeral springs) should be investigated, as it would likely further improve the reliability of the predictions.

The obtained results point out the importance of supplementing meteorological networks with hydrological monitoring and modelling for landslide hazard assessment. In fact, it sheds light on the major hydrological processes occurring in the slopes and allows identifying suitable hydrometeorological predictors of landslide occurrence. The proposed methodology, here applied to the case of pyroclastic slopes of Campania, can be replicated in other geomorphological contexts, provided that a reliable case-specific hydrological model is used.

**Data availability**

All raw data can be provided by the corresponding author upon request.

**Author contributions**

All the authors designed the research; PM and GFS developed the model code and performed the simulations; AA, DCRQ and PM analyzed the data and plotted the graphs; RG supervised the study; PM and DCRQ wrote the manuscript draft; RG reviewed and edited the manuscript.

**Competing interests**

The authors declare that they have no conflict of interest.

**Acknowledgements**

This research is part of the Ph.D. project entitled "Hydrological controls and geotechnical features affecting the triggering of shallow landslides in pyroclastic soil deposits" within the Doctoral Course "A.D.I." of Università degli Studi della Campania "L. Vanvitelli".

The research has been also funded by the Politecnico di Milano in project entitled "Multi-Risk sciEnce for resilienT commUnities undeR a changiNg climate (RETURN)", PE00000005, SPOKE VS1, CUP D43C22003030002, within the National Recovery and Resilience Plan, granted by EU - NextGenerationEU.

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
