# Peer review of "Large-scale assessment of rainfall-induced landslide hazard based on hydrometeorological information: application to Partenio Massif (Italy)"

_EGUsphere, 2024_

## Referee Comment (RC1)

**General comments:**

The manuscript presents an interesting and valuable presentation of synthetically produced meteorological data used as input for a hydro-mechanical stability model. The results demonstrate the feasibility of such model approach for landslide early warning purposes.

As the discussion is combined with the results section, a self-critical reflection of the methodology is missing. In most parts you are using an FS-model to validate the performance of the model framework, instead of relying on real landslide occurrences. When applied to real landslide data, the performance drastically decreases. A more critical discussion of the chosen approach as well as a more critical conclusion would be very necessary. Maybe, also compare the results and the model framework to other studies using synthetic datasets in landslide studies.

In my opinion, the last section (3.3) is very valuable as it applies the model for real case landslides. Here it would be interesting to present more information of the 9 landslides assessed. What are the triggering conditions (Rainfall intensity, duration etc.), what are the disposing conditions (slope angle, geotechnical parameters if available etc.). Proof/compare the parameters of large-scale application you are using (table 4) to the real case landslides. Do landslides really only occur on slopes of 40° +- 2° SD?

While the results are well presented and understandable, the manuscript lacks in quality of precise and clear English writing (misspelling, comma setting etc.). Many sentences should be rewritten making the paper more readable and easier to understand. Note, only some errors are listed below. In particular the writing of the introduction is often rocky. The instruction sometimes lacks a red thread were sentences follow one another in meaning. In my opinion, some lines should be spent on introducing the mechanical stability (FS model) and hydrological (Richard-Equation) models. Further note, that results and conclusions of your study (as in last paragraph) do not belong in introduction.

**Specific comments on lines:**

Paragraph 1&2: You are repeating "rainfall induced landslides" 3 times. Maybe state that by "landslides" you refer to "rainfall induced landslides"

Line 29: misspelling: "…it lacks a physical basis…"

Line 37: "components"

Line 43: "…increase of water stored…"

Line 42: triggering can also be related to a decrease in suction (you are also stating that in the methodlogy part).

Line 43: Achievement is weird here… Rewrite sentence.

Line 46: Instead of stating "much of the…" use "not sufficient…"

Line 61-63: Sentences not really clear.

Line 62: Weird beginning of sentence

Line 69: "per definition" makes no sense.

Line 75: Start sentence e.g. with "In this study…"

Line 86: "commonly followed" seems like an over exaggeration.

Line 87 – 92: Too long and complicated sentence, rewrite…

Line 99 – 101: Too complicated sentence, make two sentences.

Line 103 – 104: Too complicated sentence, make two sentences.

Line 105: What do you mean by "singularities"?

Line 110: "to this aim" sounds rocky…

Lien 127: "Study refers to" sounds weird.

Fig 1: Insert map of Italy is distorted. Red on green colors do not work for color blind people. Also Fig. 8 – 10 and 12.

Line 136: "travelled" is weird.

Line 137: "causing casualties" instead of "leaving human casualties".

Line 146: "was" instead of "is"

Section 2.2: Maybe rewrite title… You mostly talk about your model framework and not specifically of the synthetic dataset. Clearly state why you are using a hydrological model: to produce the 500-year synthetic dataset. Then start examining the model framework. Also note, no repetition of introduction should be made (lines e.g. 151-153).

Line 161: Fig. 2 does not show that the assumption is feasible. Maybe rewrite…

Fig 2: at the soil-bedrock interface: is this only a sink term as the direction of the arrow indicates?

Line 184: "all model parameters" instead of "all the model parameters"

Line 191: Maybe use "location" instead of "origin".

Fig. 3: Why is there such a large delay until the FS drops after H2?

Line 225: You mean Figs 3a and 3b…

Paragraph 235: Cleary state that H is the rainfall amount. Can be a bit confusing with Fig 3 as you use H1 and H2 as duration.

Line 254: "served" is weird.

Line 257-258: Is this first sentence relevant here?

Line 261: "affect" not "affects"

277: It would be nice to state at the beginning of this paragraph, why you do extensive comparison of stations: to get meteo uncertainty.

277: "rainfall amount" or something similar not "rain depth". Maybe also two sentences here, a bit rocky…

284: what is meant by "at another"?

Fig 4. Explain in caption the blue distribution line.

Lines 299 – 302: Do not repeat the same text in text and figure captions.

Lines 326: Are there seasonal differences, related to thunderstorms or similar? Did you check?

Paragraph from 327 on: State that you are considering the water content as areal means derived through ERA-5 products... Water content itself is highly variable even at small scales, surely much more than 0.02 SD (see publication on in-situ measurement). And are there even in situ water content measurements available which could be used for comparison?

Line 353: Maybe start with: "In the FS equation (5)"

Table 4: Where do you have the mean and SD values from, from the study of Roman Quitero et al. 2024? Or did you assess these values using DEM-maps and geotechnical laboratory testing? And did you test your model with other parameters? Would you get similar performances?

Line 376: I don't understand… 20 landslides in 500 years of which 10 occurred since 1999? The 10 landslides occurred in the entire region not on the reference slope. What do you mean exactly?

Line 400: "shown" not "seen"

Paragraph 424 onwards: Does that not belong to the introduction?

429: Maybe reformulate to not repeat methodology section, e.g.: "As described in section 2.4.2, Normal distributed fluctuations with standard deviation of 0.10 was introduced to account for large scale…"

Line 451: Fig. 9b does not look linear to me… What do you mean by linear equation?

Fig 9b: Why is there a drop of probability with increasing water content (at the top of the plot)?

Chapter 3.3: Did you try to run the real case landslides through the hydrological (to get the water content) and mechanical model (to get the FS)? Would be interesting to see how it performs.

Line 522: "well" not "perfectly"

Line 542: Is it really "real data" when you are not directly modelling the slopes, but just a mean and some SD?

---

## Author Comment (AC1)

**Authors' response to Referee 2 (RC2)**

On behalf of all the authors I would like to thank this Referee for taking her/his valuable time in deeply reviewing our manuscript and making valuable suggestions that will surely enrich our work. Below, you will find a point-to-point response to your comments. Your comments are presented in italics, with our responses in regular font.

*I read with interest the paper by Daniel Camilo Roman Quintero and co-authors. Their research in physically-based modelling of shallow landslides is welcome in the literature and it fits well within the scope of NHESS. They propose an approach to deal with synthetic long record of landslide occurrences and hydrometeorological conditions based on a region of Italy. Their research allow to define empirical rainfall thresholds associated with landslide occurrence in the context of landslide early warning systems. Their approach is aimed are being used on large area contexts. I also like the fact that they compare their modelling outputs to actual landslide occurrences.*

I would like to thank the Referee for the positive judgement of our work. The Referee perfectly understood the goal of our study and the methodology proposed to that aim.

*While I have limited comments over the modelling approach - the author demonstrate a strong understanding of hillslope response to rainfall and soil moisture - my main concern is, overall, that a discussion is truly missed. The authors proposed a combined results-discussion section, but apart from a link to observed landslide occurrences in the region, there is, as also pointed out by the other reviewer, not real reflection of their research with respect to the method approach, the applicability, the early warning context (to name but a few points that could be discussed) and the associated state of the art literature. To me, it makes little sense to have a manuscript without a proper discussion.*

We acknowledge that the discussion is somewhat lacking. This is likely due to our initial choice of combining the "results and discussion" in a single section, which contains a substantial amount of data and results and, therefore, obstructs a clear and extensive discussion. Therefore, in the revised paper we will separate Results from Discussion section, so to extend the discussion to better highlight potential and limitations of the proposed approach, in the context of landslide hazard assessment and early warning, and its alignment with the state of the art in the scientific literature.

*Added to the comments of the first reviewer, I have listed below other comments which I hope would be helpful to improved the manuscript:*

*The definition of "large area" is somehow unclear. 80km² for the case-study test zone is not a very large zone when compared to many landslide data-driven susceptibility assessments. There is maybe some way to better define/constrained this scale of analysis context.*

Thank you for raising this interesting issue. The distinction between large and small areas is indeed subjective and is often tied to the techniques used to understand the process under study. Data-driven susceptibility models are indeed often applied to areas larger than our study area. However, they usually rely on techniques designed to handle large datasets and extract relationships among the data without considering the physics of the involved slope processes. In our case, instead, we make use of a physically based model of the slope response to precipitation, and this can be done only if it is applied to a homogeneous geomorphological context. This is usually done with detailed flow and equilibrium models at slope scale, but here we encompass hundreds of slopes in an area of nearly 100 km$^2$. On the other side, the statistical-empirical approaches to define landslide thresholds are usually designed for larger areas, at regional or even national scale (Guzzetti et al., 2007; Peruccacci et al., 2017). However, these approaches merely rely on the correlation between rainfall and landslide occurrence, without introducing information about the physical behavior of the studied slopes.

Differently, our approach is based on a physically grounded rationale, aiming to identify causal (physical) relationships rather than mere correlations (although complex and non-linear as those that can be identified with data-driven approaches). As the Referee correctly points out, we used infiltration and slope stability models to replicate the hydrological conditions that lead to landslides in a specific region. Usually, this approach is suited to representing a specific slope at a small local scale. In our study, we started from a reference slope inclined at 40°, with a 2 m thick soil cover overlying the limestone bedrock. This geometry resembles the features of the slope involved in one of the most catastrophic landslide events in the study area: the Cervinara landslide in 1999. This event is shown with nr. 1 in Figure R2-1, where all the landslides reported in the existing catalogues are also represented. Then, we extended the analysis to a larger area, approximately 80 km², as indicated by the green-hatched region in Figure R2-1. To do this, we accounted for the effects of the variability of slope and soil properties, as well as of the effects of the variability of weather forcing.

This is the meaning of "large area" in our case: extending the results of a single slope to an area including hundreds of slopes with variable characteristics, although within the same geomorphological context. We will better clarify this aspect in the revised manuscript.

[Figure]

*Figure R2-1. Map of the large area of this study, i.e., the north-facing side of the Partenio Massif (green-filled area), with indication of the rain gauges and of the major landslides reported in available landslide catalogues from 1999 to 2020 (sourced from: (Calvello & Pecoraro, 2018; Peruccacci et al., 2023)).*

*The end of the abstract is very case-study specific. One would welcome an ending with a more general/broader statement.*

Thank you for noticing it. We will modify the statement at the end of the abstract, so to make it more general.

*Lines 33-34. To be accurate, the issue of human influence on landslides is not only in urban areas. Note also that there are some recent work (and maybe more relevant) that allow to support such a broad statement. For example, Ozturk et al. (2022)*
*https://www.nature.com/articles/d41586-022-02141-9*

We see the Reviewer's point, but our statement, probably misleading, refers to the effects of landslides on humans, and not vice versa, as the Reviewer suggests. The following sentences are also along the same line (i.e., damages and economic losses). However, we will enlarge the focus of the phrase in line 33-34 by including some references highlighting also the human influence on landslides in the revised manuscript.

*Line 35. Predicting the occurrence of landslide is also relevant when outside cities and/or when the landscape is not disturbed by human activities. Landslides are above all natural processes, and this is what the authors want to model here. The focus on urban areas from the start of the introduction is somehow misleading.*

This comment is in line with the previous one, and we agree. We will rephrase the text in the revised manuscript.

*Line 44: ..." depends not only on".. is a strange formulation. After the not only, we would expect "but also" somehow. This sentence must be rephrased.*

Thank you for catching it. We will rephrase in the revised manuscript: "landslide initiation depends not only on the triggering rainfall event characteristics, but, as in the case of shallow landslides, the achievement of instability is also favored by antecedent wet soil conditions (Mirus et al., 2018a; Wicki et al., 2020)

*Lines 75-82: it is strange to have such an emphasis on the study area in the middle of the state of the art.*

We will rephrase this part, to make clearer that, from line 75 onward, we have moved the focus from the general state-of-the-art to the literature about the specific area that we study in this paper.

*Lines 95-125: this is a very long part of the introduction to explain the goal of the research and provide a supposedly short overview of what has been done. In my opinion too many details are provided here; it sounds more like an extensive abstract.*

Thank you for suggesting it. We will reduce this final part of the introduction in the revised manuscript.

*Several times, the emphasis is put in LEWS. Although rainfall threshold determination is a key aspect of LEWS, their study can also be relevant to other hazard assessment needs. This is something that could be nuanced.*

We agree and will follow this suggestion in the revised manuscript.

*The title refers to large areas. In the introduction "wide areas" is several time used. Beyond this lack of consistency, one would appreciate a definition of what "large" actually means (size, spatial resolution, etc.)*

As described in the reply to one of the general comments from this reviewer, we will better specify the meaning of "large area" in this paper. We will also stick to the term "large"

*Figure 1. Some local names in the maps are not readable. What is the background information of the map? In such a figure one would expect a visualization of the topography to better understand, for example, the slope context.*

We don't understand what are the not readable names the Referee refers to. The smallest font size in the figure equals more or less the text font size in the caption. However, we will revise the figure to enhance its readability. Moreover, the background presents a street map dividing hillslopes (green areas) from flat areas (grey areas). We will improve the figure replacing this

background map with a topographic map with isolines to clarify the aspects outlined by the Referee.

*Line 135. Local names are being used for specific location. However, without a map to local them, such information is not relevant to a broad audience.*

If the referee is referring to the local names of the cities Rotondi, Cervinara and San Martino Valle Caudina, those locations are not only representing the municipalities hit by landslides, but also the location of the rain gauges we used to analyze rainfall spatial variability in the area. However, the locations are indicated in the map of Fig. 1. We will try to make it clearer in the revised manuscript.

*Line 138: "a huge debris avalanche". What do you mean by "huge". I would suggest not to use such a subjective wording. Here is a reference that could help:*

*McColl, S. T., & Cook, S. J. (2024). A universal size classification system for landslides. Landslides, 21(1), 111-120.*

Thank you for your suggestion. We will adopt the suggested classification in the revised manuscript, where we will also give the estimated volumes for the two mentioned landslides.

*Line 221. Replace "associated to" by "associated with"*

Thank you for catching it. We will modify the revised manuscript accordingly.

*Line 245: early warning system. Remover capital letter*

Thank you for catching it. We will modify the revised manuscript accordingly.

*Line 265. "normal…" remove capital letter*

Thank you for catching it. We will correct it in the revised manuscript.

*Lines 257-266: There seem to be some repetition here to what is being said earlier.*

Thank you for suggesting this. We recognize that there is some repetition of what is written at lines 105-111 (in the Introduction). However, this sentence here is required to properly introduce the discussion about the assessment of variability. Therefore, we will rephrase both lines 105-111 and 257-266 to minimize useless repetitions.

*Line 269: remove the two ",'*

Thank you for catching it. We will amend it in the revised manuscript.

*Figure 4 (and in other places). Is the use of decimal values in the percentage valid? Does it make sense?*

The envelopes depicted with red solid and dashed lines in Figure 4 represent the bounds containing 68.3% and 86.6% of the data, respectively. These are the percentages of the data that,

in a Normal distribution, belong to intervals centered around the mean with width of two and three standard deviations, respectively. As can be seen in many figures and tables found in literature (see an example below), it is perfectly legitimate to express percentages with decimal digits (if we misunderstood the comment, please accept our apologies).

[Figure]

*Table 2. What is "s.l.m."?*

Thank you for catching this typo. We meant m.a.s.l. (i.e., meters above sea level) and we will correct it in the revised manuscript.

*Line 229. The work of Brocca et al 2021) is mentioned as a research carried out on a relatively small area (less than 100 km²). How can we say it is a small area while, here, one speak about lar scale for an area of 80 km². As said earlier, one need a more robust definition of the scale aspect of this research.*

We think you mean line 329. Thank you again for your suggestions on this issue related to the definition of the scales (a similar issue has been also raised by the other Referee). In this study, "small scale" refers to the modeling results obtained for a single slope (the reference slope), where the variability of slope features can be neglected (or, in other words, where slope features are very well known). When we extend to an area of 80 km$^2$, it encompasses hundreds of slopes with variable geometry and soil characteristics, and this variability affects the assessment of landslide triggering. This is what we mean here with "large scale". In the paper of Brocca et al. (2012), the focus was on river catchments, and the extension of 100 km$^2$ could be considered

relatively small (by the way, we realized that, by accident, the mentioned paper by Brocca et al. (2012) was not in the reference list at the end of the manuscript, and we will fix this error in the revised manuscript). We will avoid using misleading words about this kind of scale-related concepts in the revised manuscript.

*Line 368. Remove capital letter in "Normally…"*

We think you refer to line 358. Thank you for catching this mistake. We will modify to Normal-distributed in the revised manuscript (the capital letter refers to the Normal probability distribution).

*Table 4. "large scale" instead of "Large scale"*

Thank you for catching it. We will adjust it in the revised manuscript.

*Lines 364-370. Some of this information is connected to the description of the study area. It comes a bit as a surprise that this is provided here.*

We agree that this information is connected to the description of the study area, that is mostly done in section 2.1. However, we need this information here, as it informs the uncertainty analysis, that is specifically tailored to the spatial variations observed in the study area, to assess how it affects landslide prediction. To avoid useless repetitions, we decided to report here the information written in lines 364-370. However, we will rewrite these lines, remove all the information not essential for introducing the uncertainty.

*Figure 6. Make sure that all symbols used in the figure are explained in the caption. Same comment for figure 7 and others*

Thank you for noticing it. We will adopt this suggestion in the revised manuscript.

*Line 385. Dataset instead of data set.*

Thank you for catching it. We will adjust it in the revised manuscript.

*Line 410. Such isolated sentence must be removed (or attached to a main paragraph).*

Thank you for catching it. This is just a formatting issue, that we will fix in the revised manuscript.

*Section 3.1. this is purely results, no discussion here. In sections 3.2 and 3.3 reference is only made to work that identified landslides in the region.*

Thank you for highlighting it. As mentioned before, we will separate results from discussion into two different sections in the revised manuscript, and this kind of inconsistencies will be solved.

*Line 430. 175 failures in 500 years. However, in line 376, 20 events. This is nuclear.*

Thank you for pointing out this, as it reveals that we did not make this aspect clear enough. At line 376, the analysis deals with the reference slope (local scale), without considering the variability of properties and weather forcing that one would expect moving to a larger scale. If we look at what happened to the slopes around Cervinara two landslides have been reported in 24 years (between 1999 and 2023) occurring on the same slope (small scale: smaller than one $km^2$), which is consistent with the 20 events in 500 years obtained with the synthetic dataset for the reference slope. At line 430, the spatial variability has been introduced by perturbing the dataset, thus we have moved from the reference slope to the "large area" (including all the slopes of the study area with their variable characteristics as well as the variability of the meteorological forcing). In the same 24 years, 10 landslides have been reported in the north facing side of Partenio Mountains. Interestingly, we identified 175 landslides in 500 years in the perturbed synthetic dataset, obtaining also in this case consistent rates (one landslide every 2-3 years).

We believe that these results confirm the reliability of the physically based model as well as the suitability of the synthetic data generation approach and of the way the spatial variability was accounted for. We will modify the text to better clarify all these aspects in the revised manuscript.

**References**

Brocca, L., Tullo, T., Melone, F., Moramarco, M., & Morbidelli, R. (2012). Catchment scale soil moisture spatial–temporal variability. *Journal of Hydrology*, 422-423, 63-75. https://doi.org/10.1016/j.jhydrol.2011.12.039.

Calvello, M., & Pecoraro, G. (2018). FraneItalia: a catalog of recent Italian landslides. *Geoenvironmental Disasters*, *5*(1), 1–16. https://doi.org/10.1186/S40677-018-0105-5/FIGURES/11

Guzzetti, F., Peruccacci, S., Rossi, M., & Stark, C. P. (2007). Rainfall thresholds for the initiation of landslides in central and southern Europe. *Meteorology and Atmospheric Physics*, *98*(3–4), 239–267. https://doi.org/10.1007/S00703-007-0262-7/METRICS

Peruccacci, S., Brunetti, M. T., Gariano, S. L., Melillo, M., Rossi, M., & Guzzetti, F. (2017). Rainfall thresholds for possible landslide occurrence in Italy. *Geomorphology*, *290*, 39–57. https://doi.org/10.1016/J.GEOMORPH.2017.03.031

Peruccacci, S., Gariano, S. L., Melillo, M., Solimano, M., Guzzetti, F., & Brunetti, M. T. (2023). The ITAlian rainfall-induced LandslIdes CAtalogue, an extensive and accurate spatio-temporal catalogue of rainfall-induced landslides in Italy. *Earth System Science Data*, *15*(7), 2863–2877. https://doi.org/10.5194/ESSD-15-2863-2023

---

## Author Comment (AC2)

**Authors' response to Referee 1 (RC1)**

On behalf of all the authors I would like to thank this Referee for taking her/his valuable time in deeply reviewing our manuscript and making valuable suggestions that will surely enrich our work. As follows, you will find a point-by-point response to your comments. You will find your comments in italics and our responses in normal font.

*The manuscript presents an interesting and valuable presentation of synthetically produced meteorological data used as input for a hydro-mechanical stability model. The results demonstrate the feasibility of such model approach for landslide early warning purposes.*

The referee got the point of our contribution. Thank you for the positive judgement of our work.

*As the discussion is combined with the results section, a self-critical reflection of the methodology is missing. In most parts you are using an FS-model to validate the performance of the model framework, instead of relying on real landslide occurrences.*

As noticed also by Referee #2, the choice of combining results and discussion in a single section has limited the effectiveness of the discussion, and even the presentation of the validation with real data, that we discuss in section 3.3, suffers from such a choice. Therefore, we will separate the discussion from the results in the revised manuscript, to better highlight the strong and weak points of our methodology.

As you mentioned, we presented a physically based modeling approach to estimate meteorological and hydrometeorological landslide thresholds in a landslide-prone area. To do this, we chose to rely on a synthetic dataset, rather than on historical landslides, for the following reasons. First, the number of reported historical landslides is too small to allow significant statistical analyses properly accounting for false positives, false negatives, and true alarms. Second, only the landslides that produced damage and concern are reported, and we cannot exclude that more landslides than those reported occurred (this is indeed very likely). Third, for some of the reported landslides, exact timing and location are not available, and it has not always been possible to trace them in historical imagery repositories. Indeed, lack of data is a common issue when dealing with relatively rare events like those caused by extremely unfavorable (thus unlikely) weather conditions. However, this lack of data gives the opportunity to test how synthetic data, generated thanks to physically based models, can help address data availability issues in landslide studies. This is a common approach in hydrological research, not yet well explored for landslide hazard assessment. Of course, differences between real and synthetic data are expected, even if the model we used has proven effective in replicating water drainage and slope instabilities in the past (Greco et al., 2013, 2018; Marino et al., 2021). Hence, we carried out a validation of the obtained thresholds with available real data, obtaining satisfactory results, although with some limitations.

In the revised manuscript, we will add more text to the restructured discussion to ensure that a self-critical reflection of the proposed methodology is being addressed.

*When applied to real landslide data, the performance drastically decreases. A more critical discussion of the chosen approach as well as a more critical conclusion would be very necessary. Maybe, also compare the results and the model framework to other studies using synthetic datasets in landslide studies.*

Thank you for raising this important issue. To establish thresholds using a physically based approach, the data must include the precise date, location, and preferably the material and characteristics of each landslide. For this study, we relied on the available national-scale landslide catalogs (Calvello and Pecoraro, 2018; Peruccacci et al., 2023), which, while comprehensive, unfortunately sometimes lack the spatial and temporal accuracy required for detailed studies. This could affect our model's ability to estimate triggering conditions, as some landslides may have different geomorphological origins, for instance, as well as different timing and location. These catalogs aggregate data from various sources, validated against recorded heavy rainfall events. However, other geohydrological hazards, such as rockfalls and flash floods, may also relate to observed rainfall. Despite this, the proposed thresholds successfully predict most landslides, and analyzing missed events could help understanding the decreasing in the performance.

In fact, the consulted landslide catalogs are aware of accuracy issues within the landslide data. Indeed, they label the reported events according to their accuracy. For instance, as regards temporal accuracy, FraneItalia presented by Calvello and Pecoraro (2018), classifies it in high (Td1) and low (Td2) depending on the uncertainties behind the reported date. Instead, ITALICA, presented by Peruccacci et al. (2023), classifies it in three classes: low (T3), medium (T2) and high (T1) depending on the reliable time resolution of the reports (i.e. hour, day slot and day of occurrence). Moreover, geographic accuracy is also reported. In the case of FraneItalia three classes are presented: low municipal accuracy (Sd3), medium approximate accuracy (Sd2) and high certain accuracy (Sd1). Instead, ITALICA adopts 5 classes: P0 (very high); $P1 < 1$ km2 (high); $1 \leq P10 < 10$ km2 (medium); $10 \leq P100 < 100$ km2 (low); $100 \leq P300 < 300$ km2 (very low).

Specifically, event number 3, reported on January 19, 2006, with a temporal accuracy T2 and a geographic accuracy P100, was associated with a small rainfall (28 mm recorded at the Cervinara station, 21 mm at the Rotondi station, and 33 mm at the S.M. Valle Caudina station) with relatively small intensity. The antecedent soil water content from ERA5 showed a modest value of 0.37 for the 28-100 cm layer. In the analyzed geomorphological context, these conditions are far from those typically triggering landslides (Marino et al., 2021; Sorbino et al., 2010). This event is not notable for its impacts, and there are no geomorphological markers in historical imagery, such as rupture surface scars, which are often visible years after significant landslides, leaving bare soil or changes in vegetation. In any case, there are reports of triggering meteorological alarms in the municipalities in those days.

Event number 7, reported on June 18, 2014, with a temporal accuracy T2 and a geographical accuracy P1, was linked to a rainfall event of 146 mm with an intensity of 5.4 mm/h recorded at Cervinara station. However, the antecedent soil water content recorded by ERA5 for the 28-100

cm layer was only 0.3, even below the field capacity of the pyroclastic soils in the area (Roman Quintero et al., 2023; Twarakavi et al., 2009). Documentary evidence indicates an emergency triggered by heavy rainfall throughout the area, causing primarily urban flooding in Cervinara and a small debris flow likely involving only a few cubic meters of material (UserTV Web, 2014; Zullo, 2014).

Event number 8, reported on February 8, 2019, was associated with a small rainfall event of 16 mm recorded at Cervinara station (9.2 mm at Rotondi on the same date and 43 mm at S.M. Valle Caudina two days before). The antecedent soil water content was 0.48, indicating very wet conditions, leading to the triggering of the landslide. Documentary evidence suggests that multiple landslides occurred during these days of rainfall in the area (Cronache della Campania, 2019; Nuova Irpinia, 2019).

Finally, event number 10, reported on February 15, 2020, was linked to a small rainfall event of 15 mm at Cervinara (8.6 mm and 25.3 mm at Rotondi and S.M. Valle Caudina stations) with an intensity of 2.1 mm/h. The antecedent soil water content was recorded at around 0.32 by ERA5. However, there is no documentary evidence of any morphological changes on the slopes where the landslide was reported or any hydrological emergencies around the time of the event. Figures R1-1 and R1-2 present historical satellite imagery taken on two different dates, before and after the reported occurrence of landslides 9 and 10.

[Figure]

*Figure R1-1. Satellite imagery taken in the area affected by landslide number 9 on (a) 6 August 2019 and (b) 9 May 2020.*

[Figure]

*Figure R1-2. Satellite imagery taken in the area affected by landslide number 10 on (a) 6 August 2019 and (b) 9 April 2021.*

While the occurrence of landslide 9 is well documented (see Figure R1-1) at the reported location and date, the occurrence of landslide 10 cannot be undoubtedly confirmed at the reported location and date (see Figure R1-2). If event number 10 is excluded from the threshold performance analysis, the TSS increases to approximately 0.4 and 0.6 for the meteorological and hydrometeorological thresholds, respectively.

Another critical point related to the performance with the real landslides is related to the purely meteorological threshold, which was affected by the introduction of uncertainty of rainfall in the meteorological plane (section 3.2). Specifically, in the previous version of the paper, we chose to assume no uncertainty on rainfall duration D, focusing solely on rainfall event depth H (section 2.4.1). We divided the perturbed values of H by the unperturbed durations D and obtained the mean intensities I. In this way, we implicitly assumed that the duration of rain events does not change when H changes when moving kilometers apart from a rain gauge. Differently, in the revised paper we plan to consider the perturbation of H affecting both I and D.

In line with the response to the previous comment, we will add some lines presenting a more critical discussion of the approach proposed in this manuscript, on the light of different studies, the results obtained with synthetic data and the comparison with data derived from real landslides. Specifically, we will include in the discussion some considerations on the reasons behind the apparent limited performance of the thresholds when testing them with real landslide occurrences, based on the analysis of the accuracy of the reports.

*In my opinion, the last section (3.3) is very valuable as it applies the model for real case landslides. Here it would be interesting to present more information of the 9 landslides assessed. What are the triggering conditions (Rainfall intensity, duration etc.), what are the disposing conditions (slope angle, geotechnical parameters if available etc.). Proof/compare the parameters of large-scale application you are using (table 4) to the real case landslides. Do landslides really only occur on slopes of 40° +- 2° SD?*

Thank you again for the positive feedback on this matter. We will provide additional information on the studied sample of landslides. Regarding the triggering and predisposing conditions, we associated the events with their triggering rainfall measured at the Cervinara station and the antecedent soil water content obtained from ERA5 data. Unfortunately, as mentioned in our previous response, the landslide catalogs do not always provide the exact location and timing of the events, and it has not always been possible to trace them back using satellite images. This makes it impossible to determine the precise slope inclination where some of these landslides occurred. However, it has been observed that some of the most destructive landslides occurred on slopes with an inclination ranging between 38° and 42° (Greco et al., 2021).

*While the results are well presented and understandable, the manuscript lacks in quality of precise and clear English writing (misspelling, comma setting etc.). Many sentences should be rewritten making the paper more readable and easier to understand. Note, only some errors are listed below. In particular the writing of the introduction is often rocky. The instruction sometimes lacks a red thread were sentences follow one another in meaning. In my opinion, some lines should be spent on introducing the mechanical stability (FS model) and hydrological (Richard-Equation) models. Further note, that results and conclusions of your study (as in last paragraph) do not belong in introduction.*

We sincerely thank the Referee for detailed feedback. We acknowledge the issues with English writing and the need for a smoother flow in the introduction section. We will thoroughly revise the manuscript to improve clarity, correct errors, and ensure a coherent structure. Moreover, as suggested, we will expand on the mechanical stability (FS model) and hydrological (Richards Equation) models for better context and will reduce the results and conclusions anticipated in the introduction, leaving them to the appropriate sections.

Overall, we appreciate the Referee's valuable suggestions and will implement most of these changes to enhance the manuscript's quality.

**Specific comments on lines:**

*Paragraph 1&2: You are repeating "rainfall induced landslides" 3 times. Maybe state that by "landslides" you refer to "rainfall induced landslides"*

Thank you for noticing it. We will rephrase in the revised manuscript.

*Line 29: misspelling: "...it lacks a physical basis..."*

Thank you for catching it. We will amend it in the revised manuscript.

*Line 37: "components"*

Thank you for catching it. We will amend it in the revised manuscript.

*Line 43: "...increase of water stored..."*

Thank you for seeing it. We will change it in the revised manuscript.

*Line 42: triggering can also be related to a decrease in suction (you are also stating that in the methodlogy part).*

Thank you for noticing it. We will adopt it in the revised manuscript.

*Line 43: Achievement is weird here… Rewrite sentence.*

We will change it to "attainment" in the revised version of the manuscript.

*Line 46: Instead of stating "much of the…" use "not sufficient…"*

Thank you for noticing it. We will adopt it in the revised manuscript.

*Line 61-63: Sentences not really clear.*

Thank you for catching it. We will rephrase them in the revised manuscript.

*Line 62: Weird beginning of sentence*

Thank you for catching it. We will rephrase it in the revised manuscript.

*Line 69: "per definition" makes no sense.*

Thank you for catching it. We will delete it in the revised manuscript.

*Line 75: Start sentence e.g. with "In this study…"*

Ok. We will adopt it in the revised manuscript.

*Line 86: "commonly followed" seems like an over exaggeration.*

We will rephrase it in the revised manuscript.

*Line 87 – 92: Too long and complicated sentence, rewrite…*

Thank you for noticing it. We will synthesize the idea in the revised manuscript.

*Line 99 – 101: Too complicated sentence, make two sentences.*

Ok. We will adopt this suggestion in the revised manuscript.

*Line 103 – 104: Too complicated sentence, make two sentences.*

Ok. We will adopt this suggestion in the revised manuscript.

*Line 105: What do you mean by "singularities"?*

By "singularities" we mean the many particular morphometric conditions present in the slopes, that in very detailed scale may influence the water flow processes and the slope stability, such as

subtle changes in the topography, small surface tension cracks, the presence or absence of roots, etc. We will rephrase the sentence to avoid ambiguities.

*Line 110: "to this aim" sounds rocky…*

Ok. We will amend it in the revised manuscript.

*Lien 127: "Study refers to" sounds weird.*

Thank you for catching it. We will modify it in the revised manuscript.

*Fig 1: Insert map of Italy is distorted. Red on green colors do not work for color blind people. Also Fig. 8 – 10 and 12.*

We beg to disagree with the Referee on this point. We have carefully reviewed each figure using a color blindness simulator (https://www.color-blindness.com/coblis-color-blindness-simulator/), and red on green appears to be a suitable combination for all types of color vision issues. We have also considered changing the marker shape and size to improve clarity in the maps and figures. Nevertheless, we will double-check for any 'distortion' in the map of Italy, which is intended only as a reference, and we will consider modifying it in the revised version.

*Line 136: "travelled" is weird.*

Thank you for catching it. We will amend it in the revised manuscript.

*Line 137: "causing casualties" instead of "leaving human casualties".*

Thank you for catching it. We will amend it in the revised manuscript.

*Line 146: "was" instead of "is"*

Thank you for catching it. We will amend it in the revised manuscript.

*Section 2.2: Maybe rewrite title… You mostly talk about your model framework and not specifically of the synthetic dataset. Clearly state why you are using a hydrological model: to produce the 500year synthetic dataset. Then start examining the model framework. Also note, no repetition of introduction should be made (lines e.g. 151-153).*

We agree with the Referee. We will revise the text, modify it accordingly and we will consider modifying the title into "Modelling the reference slope response to precipitation".

*Line 161: Fig. 2 does not show that the assumption is feasible. Maybe rewrite… Fig 2: at the soil-bedrock interface: is this only a sink term as the direction of the arrow indicates?*

Thank you for catching it. Fig 2 is indeed not aiming to show the feasibility of the 1D hypothesis, so the text should be amended. In fact, this figure sketches the modeled water flow processes. We will modify the text according to Referee's suggestion.

*Line 184: "all model parameters" instead of "all the model parameters"*

Thank you for catching it. We will amend it in the revised manuscript.

*Line 191: Maybe use "location" instead of "origin".*

Thank you for catching it. We will amend it in the revised manuscript.

*Fig. 3: Why is there such a large delay until the FS drops after H2?*

Fig. 3 illustrates how landslides were detected and associated to rainfall events. For clarity, the figure is a qualitative representation of various possible circumstances where unstable conditions occur within the synthetic dataset. In some cases, instability happens after a rainfall event ends, due to a delayed rainwater infiltration process.

*Line 225: You mean Figs 3a and 3b…*

Yes, thank you for catching it. We will correct it in the revised manuscript.

*Paragraph 235: Cleary state that H is the rainfall amount. Can be a bit confusing with Fig 3 as you use H1 and H2 as duration.*

Thank you for suggesting it. We will clarify this in the revised manuscript, by changing the symbols to E1, E2 and E3 to refer to the three events of Figure 3.

*Line 254: "served" is weird.*

Thank you for catching it. We will correct it in the revised manuscript.

*Line 257-258: Is this first sentence relevant here?*

We consider that the information in this phrase is relevant for the sake of overall understanding of the methodology adopted to move from the reference slope to real slopes (with uncertainty). However, we will rephrase it in the revised manuscript, to better link it with the content of the subsection.

*Line 261: "affect" not "affects"*

Thank you for catching it. We will correct it in the revised manuscript.

*277: It would be nice to state at the beginning of this paragraph, why you do extensive comparison of stations: to get meteo uncertainty.*

We will try to address it in the revised version. Thank you.

*277: "rainfall amount" or something similar not "rain depth". Maybe also two sentences here, a bit rocky…*

We will follow these suggestions in the revised manuscript.

*284: what is meant by "at another"?*

It means another station. We will add the word "station" in the revised version of the manuscript.

*Fig 4. Explain in caption the blue distribution line.*

It is an example of the assumed Normal distribution of ΔH data for a given value of $H_{med}$. It is graphically showing the correspondence of the envelopes to the 68.3% (continuous line) and 86.6% (dashed line) of the data. We will add a brief explanation in the caption of fig 4 in the revised version of the manuscript.

*Lines 299 – 302: Do not repeat the same text in text and figure captions.*

Thank you for noticing it. We will amend it in the revised manuscript.

*Lines 326: Are there seasonal differences, related to thunderstorms or similar? Did you check?*

We are aware that there should be some seasonal differences in rainfall characteristics and the hydrological response in the area (Marino et al., 2020; Pirone et al., 2015; Roman Quintero et al., 2023). Indeed, the seasonal hydrometeorological behavior may represent a further source of uncertainty to be considered. However, it is not easy to consider this issue in our kind of analysis, because, as it is well known, seasonality does not repeat exactly in the same period every year, thus it is not easy to identify the onset of summer-like or winter/spring-like hydrometeorological conditions, which may allow separating different kinds of events.

*Paragraph from 327 on: State that you are considering the water content as areal means derived through ERA-5 products...*

The referee has been probably misled by the fact that in other sections (results section 3.3) we use ERA-5 soil water content to evaluate the performance of the thresholds with real data. Here, starting from synthetic data (i.e., modeled water content) as if it was somehow known at a point (with small uncertainty), we are arguing that our choice of a SD of 0.02, that we use to perturb the data modeled for the reference slope to move to a larger scale and consider the resulting variability, would lead to reasonable results based on different sources of data (among which also ERA5).

*Water content itself is highly variable even at small scales, surely much more than 0.02 SD (see publication on in-situ measurement). And are there even in situ water content measurements available which could be used for comparison?*

We agree when the Referee states that water content is variable even at small scales. That's why we used an SD of 0.02 on $\theta$ to estimate $\Delta\theta$, which may encompass such high variability and is in line with previously published papers (Brocca et al., 2012; Dari et al., 2019). In fact, considering a normally distributed random error (refer to fig. 5a), the random fluctuations cover a range larger than 0.1, i.e., from $\Delta\theta = -0.05$ to $\Delta\theta = 0.05$. This variation, referring to the mean value of water content in the uppermost meter of pyroclastic soil deposits, is large enough to move from safe dry soil conditions to landslide-prone wet soil conditions. Unfortunately, our study area lacks specific

studies addressing the issue of spatial variability from in-situ measurements, but we recognize its importance. That's why we used a very conservative value of SD on 0.02.

*Line 353: Maybe start with: "In the FS equation (5)"*

We will include this in the revised manuscript.

*Table 4: Where do you have the mean and SD values from, from the study of Roman Quitero et al. 2024? Or did you assess these values using DEM-maps and geotechnical laboratory testing? And did you test your model with other parameters? Would you get similar performances?*

The mean and SD values come from a thorough review of the mechanical parameters of the pyroclastic ashes in the area. Actually, Roman Quintero et al. (2024) collects and compares field and laboratory data about the soil of the area, either original or taken from already published studies. We will clarify these aspects in the revised manuscript.

*Line 376: I don't understand… 20 landslides in 500 years of which 10 occurred since 1999? The 10 landslides occurred in the entire region not on the reference slope. What do you mean exactly?*

Thank you for raising this issue. This is a matter of scale. The reference slope fails approximately once every 20-25 years, according to the modelling results, and this frequency is consistent with the reported landslides in very small areas, smaller than one $km^2$, which can be considered as a single slope. Differently, slope failures across the entire 80 km² region are reported every 2-3 years, which explains the 10 landslides reported in catalogs between 1999 and 2020.

Specifically, we identified 20 landslide-triggering conditions within the unperturbed synthetic dataset covering 500 years, representative of a well-known small area near Cervinara. On average, this corresponds to one landslide every 25 years. For example, when landslide number 1 occurred (16/12/1999), landslides were triggered also on the same slope where, 20 years after, landslide number 9 happened (22/12/2019).

In contrast, the perturbed synthetic dataset, analyzed after incorporating uncertainty, displayed 175 landslide-triggering conditions over 500 years. This corresponds to a frequency aligning with field observations of a landslide occurring approximately every 2-3 years.

We will rephrase this for greater clarity in the revised manuscript.

*Line 400: "shown" not "seen"*

Thank you for catching it. We will correct it in the revised manuscript.

*Paragraph 424 onwards: Does that not belong to the introduction?*

The information in the text is relevant to the discussion, as it clarifies why spatial variability affects operational large scale hazard assessment tools. As we plan to separate the existing "Results and Discussion" section into two sections, we will determine the best way to include this information in the discussion section.

*429: Maybe reformulate to not repeat methodology section, e.g.: "As described in section 2.4.2, Normal distributed fluctuations with standard deviation of 0.10 was introduced to account for large scale..."*

Thank you for the suggestion. We will follow it in the revised manuscript.

*Line 451: Fig. 9b does not look linear to me... What do you mean by linear equation?*

We mean that the transition from low to high landslide probability can be distinguished by defining power law threshold lines in the meteorological plane and linear threshold lines in the hydrometeorological plane. We will study the way to add the envisaged lines in the probability maps to reformulate it in the revised version for clarity.

*Fig 9b: Why is there a drop of probability with increasing water content (at the top of the plot)?*

It is not very clear why the Referee mentions a 'drop in probability with increasing water content'. In Figure 9b, the probability increases as the antecedent water content increases, and it also increases with increasing rainfall. Thus, the maximum landslide probability is in the top right corner of the plot. In any case, the contour plot curves are influenced by the density of red dots, which is higher in some areas than others. We will consider changing the graph style to clarify this aspect in the revised manuscript.

*Chapter 3.3: Did you try to run the real case landslides through the hydrological (to get the water content) and mechanical model (to get the FS)? Would be interesting to see how it performs.*

We agree with the interesting suggestion by the Referee. Although the scope of this part of the study was originally to test the threshold tools in absence of modelled data, we will consider adding the comparison with modeled data. However, as mentioned before, we expect that the performance will not improve significantly, as the limited accuracy of some inventoried landslides is still determining the low performance of the thresholds.

*Line 522: "well" not "perfectly"*

We will change it in the revised manuscript.

*Line 542: Is it really "real data" when you are not directly modelling the slopes, but just a mean and some SD?*

We intended to say that we used real landslide data. We used the measured rainfall and ground antecedent conditions (i.e. volumetric soil water content) to test the thresholds, derived from synthetic data, in the reported landslide dates. We will rephrase to clarify this aspect.

**References**

Brocca, L., Tullo, T., Melone, F., Moramarco, T., & Morbidelli, R. (2012). Catchment scale soil moisture spatial–temporal variability. *Journal of Hydrology*, *422–423*, 63–75. https://doi.org/10.1016/J.JHYDROL.2011.12.039

Calvello, M., & Pecoraro, G. (2018). FraneItalia: a catalog of recent Italian landslides. *Geoenvironmental Disasters*, *5*(1), 1–16. https://doi.org/10.1186/S40677-018-0105-5

Cronache della Campania. (2019, February). *Maltempo: frana sulla statale Ofantina, chiuso lo svincolo di Parolise in Irpinia*. https://www.cronachedellacampania.it/2019/02/maltempo-frana-sulla-statale-ofantina-chiuso-lo-svincolo-di-parolise-in-irpinia/

Dari, J., Morbidelli, R., Saltalippi, C., Massari, C., & Brocca, L. (2019). Spatial-temporal variability of soil moisture: Addressing the monitoring at the catchment scale. *Journal of Hydrology*, *570*, 436–444. https://doi.org/10.1016/J.JHYDROL.2019.01.014

Greco, R., Comegna, L., Damiano, E., Guida, A., Olivares, L., & Picarelli, L. (2013). Hydrological modelling of a slope covered with shallow pyroclastic deposits from field monitoring data. *Hydrology and Earth System Sciences*, *17*(10), 4001–4013. https://doi.org/10.5194/hess-17-4001-2013

Greco, R., Comegna, L., Damiano, E., Marino, P., Olivares, L., & Santonastaso, G. F. (2021). Recurrent rainfall-induced landslides on the slopes with pyroclastic cover of Partenio Mountains (Campania, Italy): Comparison of 1999 and 2019 events. *Engineering Geology*, *288*(May 2020), 106160. https://doi.org/10.1016/j.enggeo.2021.106160

Greco, R., Marino, P., Santonastaso, G. F., & Damiano, E. (2018). Interaction between perched epikarst aquifer and unsaturated soil cover in the initiation of shallow landslides in pyroclastic soils. *Water (Switzerland)*, *10*(7). https://doi.org/10.3390/w10070948

Marino, P., Comegna, L., Damiano, E., Olivares, L., & Greco, R. (2020). Monitoring the hydrological balance of a landslide-prone slope covered by pyroclastic deposits over limestone fractured bedrock. *Water (Switzerland)*, *12*(12). https://doi.org/10.3390/w12123309

Marino, P., Santonastaso, G. F., Fan, X., & Greco, R. (2021). Prediction of shallow landslides in pyroclastic-covered slopes by coupled modeling of unsaturated and saturated groundwater flow. *Landslides*, *18*(1), 31–41. https://doi.org/10.1007/s10346-020-01484-6

Nuova Irpinia. (2019, February). *Frana sull'Ofantina bis, chiuso lo svincolo per Avellino. Percorso alternativo - Nuova Irpinia*. https://www.nuovairpinia.it/2019/02/03/frana-sullofantina-bis-chiuso-lo-svincolo-per-avellino-percorso-alternativo/

Peruccacci, S., Gariano, S. L., Melillo, M., Solimano, M., Guzzetti, F., & Brunetti, M. T. (2023). The ITAlian rainfall-induced LandslIdes CAtalogue, an extensive and accurate spatio-temporal catalogue of rainfall-induced landslides in Italy. *Earth System Science Data*, *15*(7), 2863–2877. https://doi.org/10.5194/ESSD-15-2863-2023

Pirone, M., Papa, R., Nicotera, M. V., & Urciuoli, G. (2015). Soil water balance in an unsaturated pyroclastic slope for evaluation of soil hydraulic behaviour and boundary conditions. *Journal of Hydrology*, *528*, 63–83. https://doi.org/10.1016/j.jhydrol.2015.06.005

Roman Quintero, D. C., Damiano, E., Olivares, L., & Greco, R. (2024). Mechanical and hydraulic properties of unsaturated layered pyroclastic ashes in landslide-prone areas of Campania (Italy). *Bulletin of Engineering Geology and the Environment 2024 83:7*, *83*(7), 1–12. https://doi.org/10.1007/S10064-024-03783-X

Roman Quintero, D. C., Marino, P., Santonastaso, G. F., & Greco, R. (2023). Understanding hydrologic controls of sloping soil response to precipitation through machine learning analysis applied to synthetic data. *Hydrology and Earth System Sciences*, *27*(22), 4151–4172. https://doi.org/10.5194/HESS-27-4151-2023

Sorbino, G., Sica, C., & Cascini, L. (2010). *Susceptibility analysis of shallow landslides source areas using physically based models*. 313–332. https://doi.org/10.1007/s11069-009-9431-y

Twarakavi, N. K. C., Sakai, M., & Šimůnek, J. (2009). An objective analysis of the dynamic nature of field capacity. *Water Resources Research*, *45*(10). https://doi.org/10.1029/2009WR007944

UserTV Web. (2014). *Emergenza pioggia a Cervinara, le immagini della giornata - UserTV*. https://www.youtube.com/watch?v=uYu3SXPbETY

Zullo, T. (2014, June 18). *Studio Geologico Zullo*. https://www.geologozullo.it/stampa.html